# LeMiCa: Lexicographic Minimax Path Caching for Efficient Diffusion-Based Video Generation

**Huanlin Gao**[1,2*]    **Ping Chen**[1,2*]    **Fuyuan Shi**[1,2]    **Chao Tan**[1,2]    **Zhaoxiang Liu**[1,2]
**Fang Zhao**[1,2†]    **Kai Wang**[1,2]    **Shiguo Lian**[1,2†]

Data Science & Artificial Intelligence Research Institute, China Unicom[1]
Unicom Data Intelligence, China Unicom[2]

{gaohl51, chenp181, shify15, tanc10, liuzx178, zhaof50, wangk115,
liansg}@chinaunicom.cn

https://unicomai.github.io/LeMiCa

## Abstract

We present LeMiCa, a training-free and efficient acceleration framework for diffusion-based video generation. While existing caching strategies primarily focus on reducing local heuristic errors, they often overlook the accumulation of global errors, leading to noticeable content degradation between accelerated and original videos. To address this issue, we formulate cache scheduling as a directed graph with error-weighted edges and introduce a Lexicographic Minimax Path Optimization strategy that explicitly bounds the worst-case path error. This approach substantially improves the consistency of global content and style across generated frames. Extensive experiments on multiple text-to-video benchmarks demonstrate that LeMiCa delivers dual improvements in both inference speed and generation quality. Notably, our method achieves a 2.9× speedup on the Latte model and reaches an LPIPS score of 0.05 on Open-Sora, outperforming prior caching techniques. Importantly, these gains come with minimal perceptual quality degradation, making LeMiCa a robust and generalizable paradigm for accelerating diffusion-based video generation. We believe this approach can serve as a strong foundation for future research on efficient and reliable video synthesis. Our code is available at https://github.com/UnicomAI/LeMiCa

## 1  Introduction

Diffusion models [10, 38] have made significant advancements in video generation [24, 53, 45], particularly with DiT-based architectures [29], which greatly enhance visual quality. However, these methods are often hindered by high memory usage, substantial computational costs, and long inference latencies, limiting their use in interactive applications. This has led to increased interest in more efficient and cost-effective generation strategies.

Existing approaches such as model distillation [39, 30, 42], pruning [7, 27], and quantization [34, 37, 8, 18] have been widely adopted to accelerate inference. While effective, these methods require careful architectural design and retraining on large datasets, incurring high costs. Caching mechanisms [35, 26], in contrast, offer a retraining-free alternative for accelerating diffusion model inference. The core idea is to reuse model outputs from specific timesteps during sampling to reduce redundant computations and speed up the process [20, 44]. Selecting optimal cache timesteps, while balancing video quality and inference speed, remains an open problem in video generation.

Ideally, a lossless video acceleration method should meet two essential criteria: **(i) High visual quality** and **(ii) Consistency between accelerated and original videos**. However, existing cache-

---

[*]Equal contribution
[†]Corresponding author

based methods [20, 51] maintain a certain level of visual quality, but they often introduce content deviations and loss of high-frequency details, increasing the risk of uncontrolled degradation.

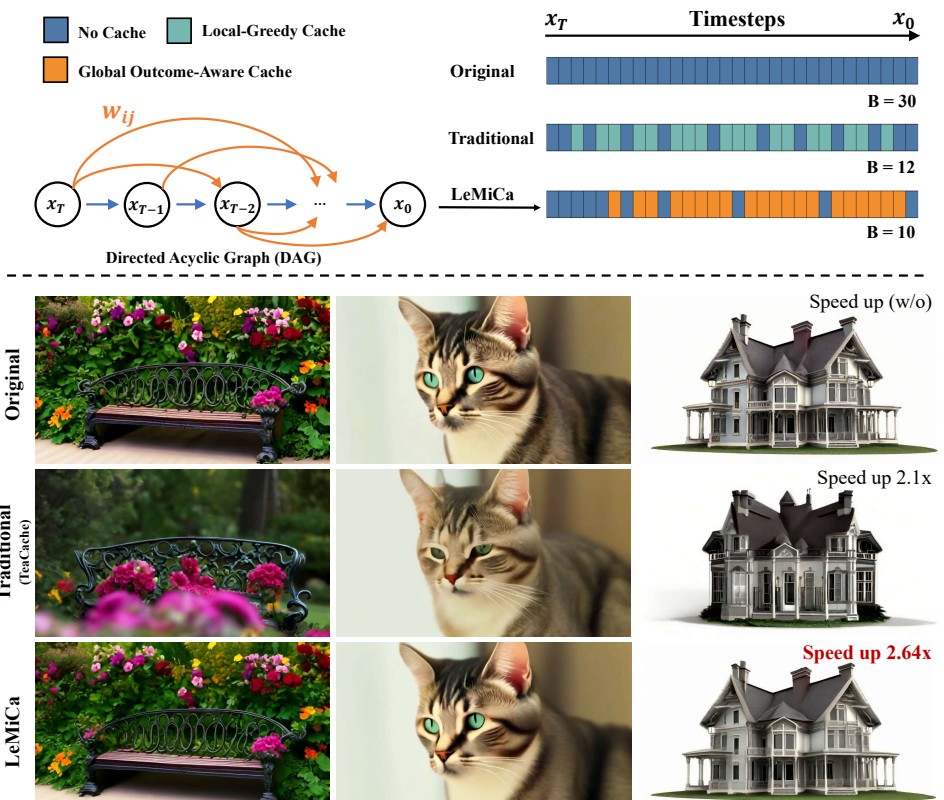

Figure 1: Comparison between our globally controlled cache mechanism (**LeMiCa**) and traditional local greedy cache methods. **Top**: The second row shows the traditional Local-Greedy approach, which uses local error estimation and fixed thresholds for caching decisions. It assumes uniform denoising contributions across time steps and ignores temporal heterogeneity and error propagation. Our method (third row) introduces a *Global Outcome-Aware Cache*, evaluating cache segment impacts through multiple prompts along a fixed sampling path, creating a static directed acyclic graph (DAG). We then use *Lexicographic MiniMax Path Optimization* (LeMiCa) to find the optimal cache path under a fixed inference budget ($B$, model forward steps). **Bottom**: LeMiCa outperforms traditional methods (e.g., TeaCache) in maintaining structural consistency with faster inference and better control over cache errors and distortions.

Upon further analysis, we identify two key limitations. First, representative methods [44, 20] typically compute local errors between adjacent timesteps and apply fixed thresholds to decide whether to cache. However, the diffusion denoising process exhibits significant *temporal heterogeneity*, with varying noise levels and semantic richness across timesteps. Applying a uniform threshold throughout the process may disrupt semantic alignment and introduce inconsistencies in decision-making, leading to inaccurate caching behavior. Second, these methods mainly focus on minimizing local differences between consecutive steps—what we refer to as *Local-Greedy error*. While this may reduce short-term discrepancies, it overlooks how small errors accumulate over time, potentially resulting in a dual loss in both video quality and content consistency. These issues are evident in TeaCache (a state-of-the-art Local-Greedy method), as shown in Figure 1, particularly with the three frames in the top and second rows, where caching introduces noticeable content deviations and visual quality degradation.

To address these limitations, we propose **Le**xicographic **Mi**nimax **Ca**ching (**LeMiCa**), a static caching framework that is model-agnostic and architecture-independent. Instead of using local greedy

strategies, LeMiCa treats cache scheduling as a global path planning problem. This is based on the observation that well-trained diffusion models remain stable along a fixed sampling path.

LeMiCa takes a global view of error by introducing the *Global Outcome-Aware error*, which quantifies the impact of each cache segment on the final output, effectively eliminating temporal heterogeneity and mitigating error propagation. Based on this metric, LeMiCa constructs a *Directed Acyclic Graph* (DAG), where each edge represents a possible cache segment and is weighted by its global impact on output quality. This graph is generated offline using multiple prompts and full sampling trajectories.

We then apply *lexicographic minimax optimization* to identify the path that minimizes worst-case degradation. Among all feasible paths under a fixed budget, the one with the smallest maximum error is selected. If multiple paths have the same maximum error, the next largest error is compared, and so on. This strategy explicitly constrains the worst-case error, effectively preventing global degradation caused by locally unstable cache decisions, and significantly improving content consistency and video quality in accelerated generation.

In summary, the contributions of this paper are:

- We propose **LeMiCa**, a novel, training-free cache scheduling framework that formulates the generation process as a globally optimized DAG traversal task, offering a principled alternative to heuristic and locally greedy approaches.

- We conduct an in-depth analysis of the cache optimization problem and appropriately introduce the Lexicographic Minimax Path Optimization strategy to solve the graph under a fixed cache budget, effectively suppressing error peaks and enhancing global consistency.

- Experiments show that, compared to existing cache techniques, ours achieves dual improvements in inference speed and generation quality across various base models, such as a 2.9X speedup on Latte and an LPIPS of 0.05 on Open-Sora.

## 2  Related Work

**Diffusion Model Acceleration.**    Diffusion models exhibit strong versatility across domains, but their iterative nature incurs high computational costs, positioning inference acceleration as a central research challenge. Current efforts to accelerate diffusion model sampling focus primarily on reducing sampling steps via schedulers. Denoising Diffusion Implicit Models (DDIM) [38] represents one of the earliest attempts to accelerate sampling by extending the original Denoising Diffusion Probabilistic Model (DDPM) [10] to non-Markovian settings. The Efficient Denoising Model (EDM) [13] introduces a design framework that optimizes specific aspects of the diffusion process. Concurrently, there is growing attention to more efficient and accurate methods for solving stochastic differential equations (SDEs) and ordinary differential equations (ODEs) [40, 12, 21, 3]. Other approaches introduce knowledge distillation [9], training a student model to condense the multi-step outputs of the original diffusion model into fewer steps [22], including Progressive Distillation [30], Consistency Distillation [39, 14, 6, 42, 52], Adversarial Diffusion Distillation [32, 31], and Score Distillation Sampling [47, 46]. Additionally, methods such as quantization [17, 36, 34], pruning [7, 27], optimization [19], and parallelism [50, 15, 5, 4] have been proposed and applied to various diffusion-based generative tasks. However, these methods often require large amounts of computational resources and data for training or intricate engineering designs, which increases the complexity of their application.

**Cache in Diffusion Models.**    Caching mechanisms [35] have recently attracted attention as a retraining-free alternative for accelerating diffusion model inference [44, 25]. The core idea is to reuse model outputs from certain timesteps during sampling to reduce redundant computations [33]. DeepCache [26] accelerates the Unet structure using manually set rules. T-GATE [49] and $\Delta$-DiT [2] apply this idea to DiT-based networks [29], achieving advanced image generation acceleration [54, 16]. With the breakthrough of Sora [28] in video generation, researchers have extended this acceleration concept from image generation to video generation. In this context, PAB [51] observed a U-shaped pattern in attention differences across timesteps in the diffusion process, and based on this, proposed a strategy to cache and broadcast intermediate features at various timestep intervals. FasterCache [23] realized the significant redundancy in conditional generation (CFG) and further enhanced inference speed by utilizing a dynamic feature-based caching mechanism. TeaCache [20] leverages the correlation between timestep embeddings and model outputs, incorporating threshold-based indicators

and polynomial fitting to guide caching. Although these methods have improved the efficiency of diffusion-based generation, the core challenge remains in how to accelerate inference while maintaining content consistency and preserving details.

## 3    Method

### 3.1    Background: Denoising Diffusion Models

Denoising Diffusion Models achieve generative modeling by simulating the gradual noising and denoising process of data. The core of these models consists of two key stages: **diffusion** and **denoising**. During the forward diffusion process, the model starts from a real sample $x_0 \sim q(x)$ and gradually adds Gaussian noise over $T$ timesteps. The noised sample $x_t$ at timestep $t$ is given by:

$$x_t = \sqrt{\alpha_t}\, x_{t-1} + \sqrt{1 - \alpha_t}\, z_t, \quad z_t \sim \mathcal{N}(0, I), \quad t = 1, \ldots, T, \tag{1}$$

where $\alpha_t$ controls the noise strength at each step. As $t$ increases, the samples converge to a standard normal distribution $\mathcal{N}(0, I)$. In the reverse denoising process, the model reconstructs the original data distribution by iteratively denoising through a neural network. The conditional probability for each step is modeled as:

$$p_\theta(x_{t-1} \mid x_t) = \mathcal{N}(x_{t-1}; \mu_\theta(x_t, t), \Sigma_\theta(x_t, t)), \tag{2}$$

where $\mu_\theta$ and $\Sigma_\theta$ are learned mean and covariance functions. Due to the multi-step nature of denoising, diffusion models typically incur significant computational overhead during generation.

### 3.2    Rethinking Cache in Diffusion Sampling

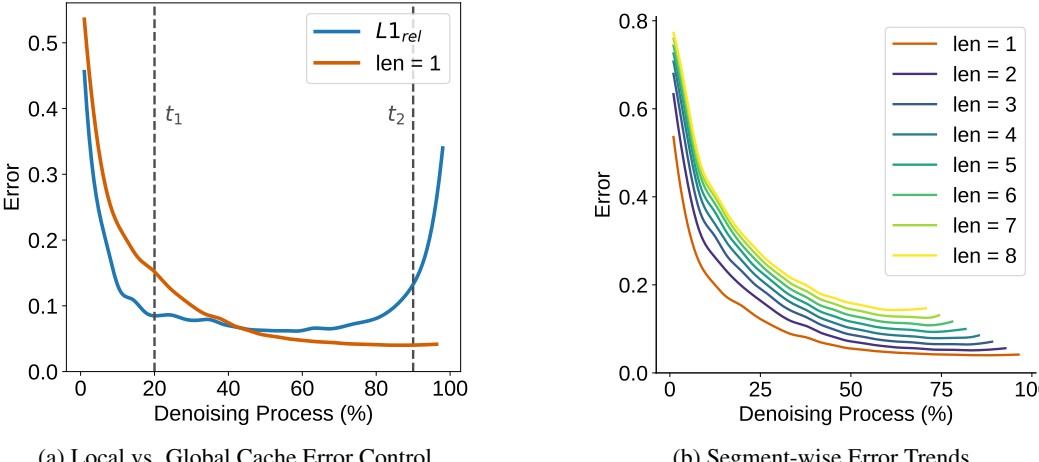

(a) Local vs. Global Cache Error Control

(b) Segment-wise Error Trends

Figure 2: Rethinking cache reuse in denoising diffusion via error estimation. (a) The traditional Local-Greedy ($L1_{rel}$) strategy uses fixed thresholds on local output differences between adjacent timesteps to decide when to cache. This assumes uniform temporal sensitivity, which can be misleading—for instance, caching at $t_2$ yields lower final error than $t_1$, despite $t_1$ seeming smoother locally. This highlights the role of temporal heterogeneity. (b) Our Global Outcome-Aware (*segment-wise error*) strategy estimates final output error when caching outputs over segments of length $len$, starting from timestep $i$. The plot shows that early caches cause greater error, supporting an outcome-sensitive, trajectory-aware strategy over fixed local heuristics.

Traditional cache reuse in diffusion sampling typically adopts a *Local-Greedy* strategy (Figure 2a), where caching is based on local differences between adjacent model outputs, often measured by the relative L1 distance [20]:

$$\mathrm{L1}_{\mathrm{rel}}(O, t) = \frac{\|O_t - O_{t+1}\|_1}{\|O_{t+1}\|_1} \tag{3}$$

where $O_t$ is the output at timestep $t$. High local differences prompt full inference; low differences lead to cache reuse. This step-wise strategy assumes uniform importance across timesteps.

However, diffusion processes are inherently *temporally heterogeneous*—early steps shape global structure, while later steps refine details. Thus, as illustrated in Figure 2a, a seemingly minor change at an early step (e.g., $t_1$) can have a larger impact on the final output than a larger change at a later step (e.g., $t_2$). Local metrics fail to account for this asymmetric error propagation, motivating a rethinking of cache strategies.

To address this, we propose a *Global Outcome-Aware* view that considers the *long-term impact* of cache reuse over time. Specifically, we define a *cache segment* $(i, j)$ means full inference is performed at timesteps $i$ and $j$, while all intermediate steps $t \in (i, j)$ reuse cached outputs:

$$\text{L1}_{\text{glob}}(i \to j) = \frac{1}{N} \left\| x_0^{\text{cache}(i \to j)} - x_0^{\text{original}} \right\|_1 \tag{4}$$

Here, $x_0^{\text{original}}$ is the output with no caching, and $x_0^{\text{cache}(i \to j)}$ is the output with segment-level cache. As shown in Figure 2b, the global error depends not just on segment length but also on its *temporal position*—early caches induce amplified downstream errors, while later caches are less disruptive.

These findings reveal two key insights: (1) Global error propagation is *non-uniform* and *time-dependent*, invalidating fixed-threshold heuristics; (2) The *position* of the cache segment matters more than its length. Building on these insights, we formulate cache planning as a graph-based constrained path optimization problem over the sampling trajectory.

### 3.3 Lexicographic Minimax Path Caching

Based on the rethinking of cache in Sec 3.2, we propose **LeMiCa**, a method that integrates sparse directed graph construction with optimal graph search under peak error control.

**Graph Construction.** We construct a directed acyclic graph, as shown in Figure 1, where each edge represents a candidate cache segment along the original sampling trajectory. To reduce complexity, we impose a maximum skip length based on the prior that long-range reuse typically leads to large errors, thus avoiding full graph construction. Edge weights are evaluated by replaying cached segments using intermediate states from a full denoising pass. To ensure generality, we build a static graph by averaging edge errors across diverse prompts and noise seeds.

**Graph Optimization.** Given a directed acyclic graph $G$ with globally error-weighted edges, we frame the caching problem as selecting a path from source $s$ to target $t$ that includes exactly $B$ full computation steps and an arbitrary number of cached segments. This budget-constrained formulation allows flexible reuse while bounding computational cost.

As shown in Figure 2b, early-stage cache errors amplify exponentially during denoising, while late-stage errors remain more localized. This asymmetric error propagation renders traditional shortest-path heuristics— which minimize only additive cost—suboptimal, as they fail to control the dominant sources of degradation.

To better address this imbalance, we adopt a *lexicographic minimax* criterion that explicitly minimizes the highest cache error along the path, followed by the second highest, and so on. Unlike training-based approaches such as ShortDF [3], which directly seek the shortest error path, our formulation—commonly used in control systems for robust worst-case optimization—offers improved stability in error-sensitive settings. Formally, the optimization problem is defined as:

$$\min_{P \in \mathcal{P}_{s \to t}^{(B)}} \text{LexMax} \left( \text{sort\_desc} \left( \{ w(e) \mid e \in P_{\text{cache}} \} \right) \right) \tag{5}$$

Here, $\mathcal{P}_{s \to t}^{(B)}$ denotes the set of all paths from $s$ to $t$ with exactly $B$ full steps, and $P_{\text{cache}} \subset P$ are the cached segments within a given path $P$. The operator LexMax lexicographically minimizes the sorted error vector, ensuring worst-case robustness. The detailed algorithm pseudocode is provided in the Appendix (Section A).

Table 1: Comparison of inference efficiency and visual quality across different models and acceleration strategies on a single GPU.

| Method | Efficiency | | | Visual Quality | | | |
|---|---|---|---|---|---|---|---|
| | FLOPs (P)↓ | Speedup↑ | Latency (s)↓ | VBench↑ | LPIPS↓ | SSIM↑ | PSNR↑ |
| **Open-Sora 1.2 (51 frames, 480P)** | | | | | | | |
| Original | 3.15 | 1× | 26.54 | 79.22% | — | — | — |
| Δ-DiT | 3.09 | 1.03× | 25.87 | 78.21% | 0.569 | 0.481 | 11.91 |
| T-GATE | 2.75 | 1.19× | 22.22 | 77.61% | 0.350 | 0.676 | 15.50 |
| PAB | 2.50 | 1.43× | 18.52 | 76.95% | 0.174 | 0.822 | 23.58 |
| TeaCache-slow | 2.40 | 1.50× | 17.58 | 79.20% | 0.134 | 0.837 | 23.50 |
| TeaCache-fast | 1.64 | 2.10× | 12.63 | 78.24% | 0.252 | 0.743 | 19.03 |
| LeMiCa-slow | 2.30 | 1.52× | 17.43 | **79.26%** | **0.050** | **0.923** | **31.32** |
| LeMiCa-fast | **1.45** | **2.44×** | **10.86** | 78.34% | 0.187 | 0.798 | 21.76 |
| **Latte (16 frames, 512×512)** | | | | | | | |
| Original | 3.36 | 1× | 11.18 | 77.40% | — | — | — |
| Δ-DiT | 3.36 | 1.02× | 10.85 | 52.00% | 0.851 | 0.108 | 8.65 |
| T-GATE | 2.99 | 1.13× | 9.88 | 75.42% | 0.261 | 0.693 | 19.55 |
| PAB | 2.52 | 1.36× | 8.21 | 73.13% | 0.390 | 0.642 | 17.16 |
| TeaCache-slow | 1.94 | 1.65× | 6.76 | 77.40% | 0.195 | 0.775 | 21.52 |
| TeaCache-fast | 1.15 | 2.60× | 4.30 | 76.09% | 0.318 | 0.674 | 18.04 |
| LeMiCa-slow | 1.88 | 1.69× | 6.60 | **77.45%** | **0.091** | **0.865** | **27.65** |
| LeMiCa-fast | **1.00** | **2.93×** | **3.81** | 76.75% | 0.273 | 0.70 | 19.43 |
| **CogVideoX (49 frames, 480P)** | | | | | | | |
| Original | 12.45 | 1× | 43.08 | 77.13% | — | — | — |
| PAB | 9.26 | 1.43× | 32.07 | 75.95% | 0.064 | 0.916 | 29.85 |
| TeaCache-slow | 6.93 | 1.70× | 25.34 | 76.79% | 0.053 | 0.928 | 31.07 |
| TeaCache-fast | 4.53 | 2.45× | 17.58 | 76.06% | 0.176 | 0.804 | 22.95 |
| LeMiCa-slow | 6.91 | 1.72× | 25.02 | **76.89%** | **0.023** | **0.958** | **35.93** |
| LeMiCa-fast | **4.26** | **2.61×** | **16.48** | 76.20% | 0.132 | 0.846 | 25.59 |

# 4 Experiments

## 4.1 Experimental Setup

**Metrics** For fair comparison, we follow prior works and report both efficiency and visual quality metrics. Efficiency is measured by FLOPs and latency. Visual quality is evaluated using VBench [11] (human preference), LPIPS [48] (perceptual similarity), SSIM [43] (structural consistency), and PSNR (pixel-level accuracy).

**Baselines and Compared Methods** We evaluate our method on representative diffusion-based video models: Open-Sora [53], Latte [24], and CogVideoX [45]. Baselines include Δ-DiT [2], T-GATE [49], PAB [51], and TeaCache [20]. Among them, T-GATE and Δ-DiT are designed for images, while PAB and TeaCache target video. Accordingly, we compare against PAB and TeaCache on CogVideoX, and against all four baselines on Open-Sora and Latte.

**Implementation Details** Experiments are conducted on NVIDIA H100 GPUs using PyTorch. To construct the DAG for Global Outcome-Aware error modeling, we sample 70 prompts (10 per attribute) from T2V-CompBench [41], following standard practice [41, 20]. The DAG construction and forward inference use distinct datasets to ensure fair and robust evaluation. Sampling is repeated 10 times with different seeds, and results are averaged to reduce bias.

## 4.2 Comparison with State-of-the-Art Methods

**Quantitative Comparison** Table 1 compares LeMiCa with baselines across four metrics: VBench, LPIPS, SSIM, and PSNR. LeMiCa includes two variants: LeMiCa-slow (fidelity-focused) and

LeMiCa-fast (speed-focused). It consistently outperforms training-free acceleration baselines across models, schedulers, resolutions, and video lengths. LeMiCa-slow achieves the best reconstruction quality, reducing LPIPS from 0.134 to 0.05 on Open-Sora and from 0.195 to 0.091 on Latte—over 2× improvement vs. TeaCache-slow. LeMiCa-fast improves inference speed from 2.60× to 2.93× on Latte compared to TeaCache-fast, while preserving visual quality. Unlike prior methods relying on online greedy strategies, LeMiCa precomputes its caching policy, eliminating runtime overhead. Overall, LeMiCa provides efficient video generation with minimal perceptual quality degradation.

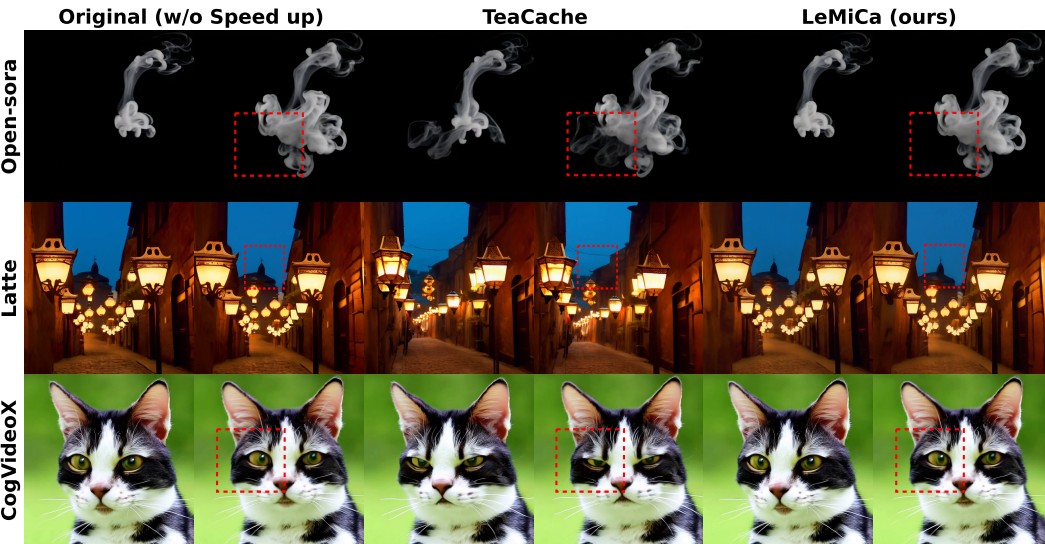

Figure 3: Visual comparison under the fidelity-focused setting (LeMiCa-slow vs. TeaCache-slow) across different models. Differences are highlighted in red boxes.

**Visualization**    We compare video acceleration methods from both quality and speed perspectives. As shown in Fig. 3, under the fidelity-focused setting, LeMiCa excels in preserving content consistency and fine details, as highlighted in red boxes. This demonstrates its ability to maintain high-quality visuals even when prioritizing fidelity. In contrast, Fig. 4 illustrates that under the speed-focused se-

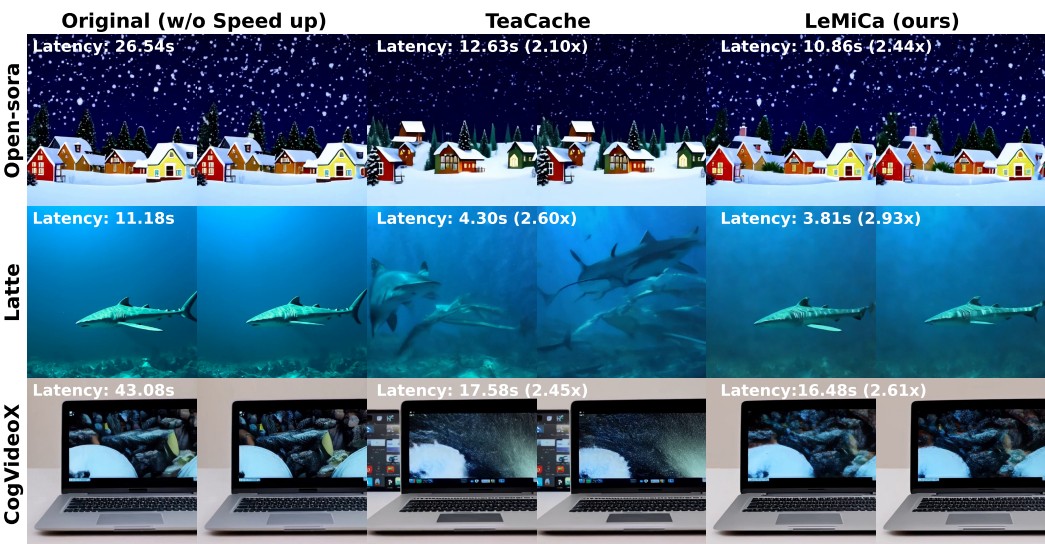

Figure 4: Visual comparison under a speed-focused setting (LeMiCa-fast vs. TeaCache-fast). LeMiCa-fast better preserves content consistency and video quality under a high speedup (>2×).

tting, LeMiCa-fast significantly outperforms TeaCache-fast, achieving superior acceleration rates while still maintaining competitive performance. These results highlight LeMiCa's ability to balance quality and speed across different configurations. Additional qualitative examples can be found in the Appendix (Section E).

### 4.3 Ablation Studies

**Acceleration vs. Performance trade-off**   Figure 5 presents the quality-latency trade-off between our proposed LeMiCa and TeaCache. To ensure comparable computational budgets, LeMiCa is configured with 19, 12, 9, and 7 inference steps (i.e., inference budget $B$), corresponding to TeaCache thresholds of 0.1, 0.2, 0.3, and 0.5, respectively. Across all latency regimes, LeMiCa consistently achieves a superior quality-efficiency balance, outperforming TeaCache on all reference-based metrics. Importantly, under extreme acceleration (latencies below 8 seconds), LeMiCa maintains robust and high-quality performance.

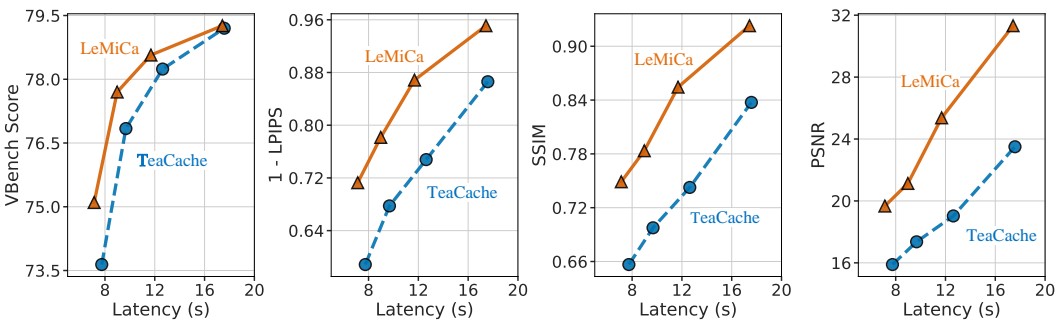

Figure 5: Quality-latency trade-off comparison between LeMiCa and TeaCache.

**Sample Requirements for Graph Construction**   To investigate how many samples LeMiCa requires to offline construct the DAG, we randomly select $n \in \{1, 5, 10, 20\}$ from the original 350 samples (70 prompts × 5 seeds), and compute the optimal caching path under the lexicographic minimax criterion. Each setting is repeated 20 times to reduce randomness. Importantly, distinct datasets are used for DAG construction and forward inference to guarantee fairness and robustness in evaluation. Video quality is then evaluated on 50 selected VBench prompts, with average results reported. Table 2 shows that LeMiCa achieves strong performance with a single sample (e.g., PSNR 24.51), rapidly approaching the upper bound with 10 samples and essentially saturating at 20 samples across all metrics. This demonstrates LeMiCa's ability to construct high-quality cache paths with minimal samples and underscores the robustness of its static caching strategy across varying prompts and seeds.

Table 2: Impact of sample size on cache path graph quality.

| Number of Samples | VBench↑ | LPIPS↓ | SSIM↑ | PSNR↑ |
|---|---|---|---|---|
| 1 | 78.58 | 0.164 | 0.838 | 24.51 |
| 5 | 78.70 | 0.161 | 0.843 | 24.57 |
| 10 | 78.95 | 0.158 | 0.844 | 24.56 |
| 20 | 79.16 | 0.152 | 0.843 | 24.60 |
| 350 | 79.27 | 0.143 | 0.851 | 24.67 |

Table 3: Impact of different path strategies on video reconstruction quality.

| Path Strategy | VBench↑ | LPIPS↓ | SSIM↑ | PSNR↑ |
|---|---|---|---|---|
| Original | 79.24 | - | - | - |
| Shortest Path | 76.04 | 0.203 | 0.809 | 22.90 |
| MiniMax Path | 79.27 | 0.143 | 0.851 | 24.67 |

**Trajectory Robustness**   Since the cache mechanism is inherently tied to the original denoising trajectory, it is essential to assess whether a training-free cache method remains effective when the trajectory changes. To this end, we vary the trajectory scale parameter in the sampling schedule from its default value of 1.0 to several alternative values (0.5, 0.75, 1.25, 1.5), introducing different diffusion paths during inference. As shown in Figure 6, the left panel illustrates the effect of trajectory scaling on the denoising paths, while the right panel demonstrates that LeMiCa consistently outperforms the current state-of-the-art method, TeaCache, across all trajectories in terms of LPIPS. These results confirm that our method remains effective even under varying denoising paths.

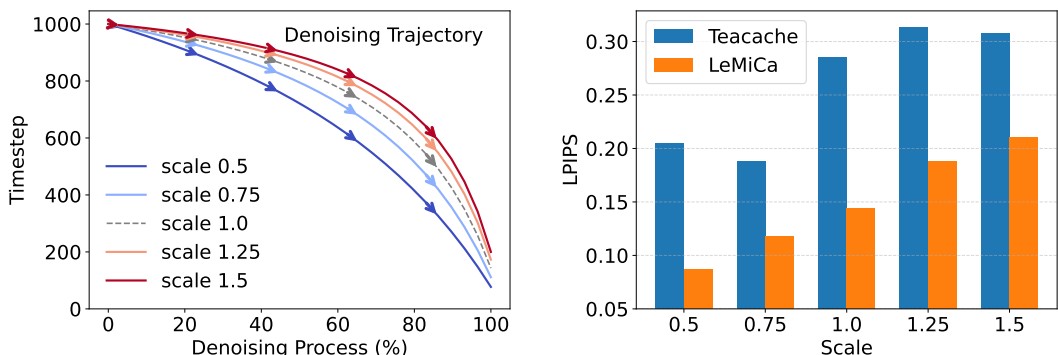

Figure 6: Performance comparison between LeMiCa and TeaCache across different denoising trajectories. Left: Denoising step trajectories under different scale settings. Right: LPIPS performance across various denoising trajectories.

**Shortest Path vs. Lexicographic MiniMax Path**  We compare the performance of the *Shortest Path* strategy and the *Lexicographic MiniMax Path* strategy in video reconstruction tasks. As shown in Table 3, the *MiniMax Path* strategy consistently outperforms the baseline *Shortest Path* strategy in both VBench scores and reconstruction metrics. This observation is consistent with our analysis: the edge errors cached during the sampling process are not independent and thus cannot be simply accumulated linearly.

**Performance at different resolutions and lengths**  Our method incorporates Dynamic Sequence Parallelism (DSP) [51]to support high-resolution long-video generation across multiple GPUs. To assess its sampling acceleration performance across varying video sizes, we conducted tests on videos with different lengths and resolutions. As shown in Figure 7, our method maintains stable acceleration even as video resolution and frame count increase, highlighting its potential for handling longer and higher-resolution videos.

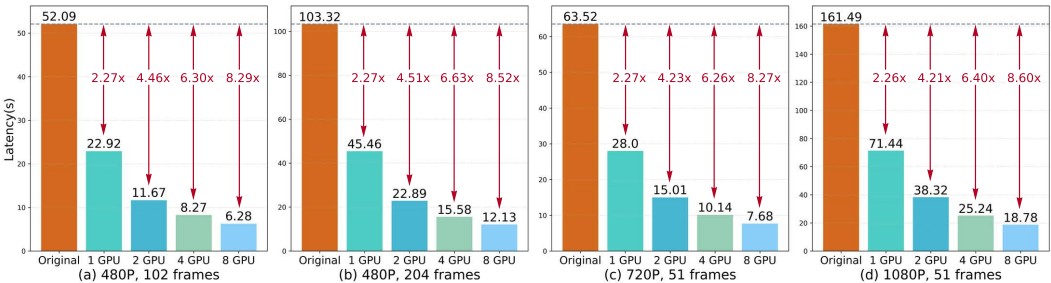

Figure 7: LeMiCa inference efficiency under various video durations and resolutions.

# 5   Conclusion

We propose **LeMiCa**, a general and efficient caching framework for accelerating diffusion-based video generation. Unlike locally greedy strategies, LeMiCa formulates cache scheduling as a global path optimization problem using lexicographic minimax over a static DAG, effectively constraining worst-case degradation. With the introduction of the Global Outcome-Aware error, our method captures the long-term impact of caching decisions, mitigating temporal heterogeneity and error accumulation. Extensive experiments demonstrate that LeMiCa consistently improves both efficiency and visual quality across diverse diffusion models. More broadly, LeMiCa offers a new perspective on structured caching in generative modeling, which may inspire future research in other domains such as 3D, multi-view, or multi-modal generation where controllable acceleration remains an open challenge.

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

# LeMiCa: Lexicographic Minimax Path Caching for Efficient Diffusion-Based Video Generation

# Appendix

## A  Pseudocode: Lexicographic Minimax Path Selection

Below we present the full pseudocode implementation of the lexicographic minimax path selection algorithm, as introduced in Section 3.3. The algorithm leverages dynamic programming to efficiently compute an optimal caching path from source $s$ to target $t$ under a step budget constraint $B$. Unlike standard shortest-path methods, our approach handles the non-additive and non-Markovian nature of error accumulation by minimizing edge weights in a lexicographically ordered manner.

---
**Algorithm 1** Lexicographic Minimax Path Selection
---
1: **Input:** Directed acyclic graph $G = (V, E)$, start node $s$, end node $t$, step limit $B$
2: **Output:** Lexicographic Minimax Path $P^*$
3: **Initialization:**
4:    $dp[v][k]$: maximum edge weight on any $k$-step path to $v$
5:    $paths[v][k], edges[v][k]$: corresponding node and edge sequences
6:    $dp[s][0] \leftarrow 0, paths[s][0] \leftarrow [[s]], edges[s][0] \leftarrow [[]]$
7: **Main Loop:**
8: **for** $k = 0$ to $B - 1$ **do**
9:    **for** each node $v$ with $dp[v][k] < \infty$ **do**
10:       **for** each neighbor $u$ of $v$ **do**
11:          $w \leftarrow$ weight of edge $(v, u)$
12:          $m \leftarrow \max(dp[v][k], w)$
13:          **if** $m < dp[u][k + 1]$ **then**
14:             $dp[u][k + 1] \leftarrow m$
15:             Update $paths[u][k + 1], edges[u][k + 1]$ from $v$
16:          **else if** $m = dp[u][k + 1]$ **then**
17:             Append new paths and edges from $v$ to $paths[u][k + 1], edges[u][k + 1]$
18:          **end if**
19:       **end for**
20:    **end for**
21: **end for**
22: **Final Selection:**
23: $P^* \leftarrow \min\big(\text{zip}(paths[t][B], edges[t][B]), \text{key} = \lambda(p, e) : \text{sorted}(e, \text{reverse=True})\big)$
---

## B  Experiment Settings

### B.1  Models

In this paper, we introduce LeMiCa, a novel caching technique designed to accelerate and enhance a range of state-of-the-art video synthesis models, including Open-Sora 1.2 [53], Latte [24], and CogVideoX [45]. Open-Sora 1.2 integrates 2D/3D VAEs and ST-DiT blocks for efficient video compression and generation. Latte leverages spatio-temporal tokenization and Transformer layers to model video distributions in the latent space. CogVideoX employs a 3D VAE and expert Transformers with adaptive LayerNorm for modality fusion and high-fidelity generation. In our experiments, we adopt the CogVideoX-2B variant.

### B.2  Details of the Compared Methods

**PAB** introduces a pyramid-style broadcasting mechanism to reduce redundant attention computations in diffusion models. By observing a U-shaped pattern in attention differences across steps, PAB

applies adaptive broadcast strategies based on the variance of different attention types (e.g., spatial, temporal, cross-modal). Stable attention outputs are efficiently reused in later steps, reducing computation. All experiments use PAB's default parameter settings.

**TeaCache** is a training-free, architecture-agnostic caching method that exploits the correlation between timestep embedding changes and model output differences across adjacent steps. By introducing a unified threshold-based strategy, TeaCache decides when to activate caching through an accumulated error-based discriminator. Since this method operates solely along the temporal dimension without modifying specific model components, it offers strong generalization and broad applicability.

### B.3 Model Forward Steps

**Model Forward Steps.** In this work, we control the acceleration efficiency of LeMiCa via the Model Forward Steps $B$. Smaller values of $B$ reduce the denoising time, leading to higher speed-up ratios. We consider two variants: LeMiCa-slow, which emphasizes visual fidelity, and LeMiCa-fast, which prioritizes inference efficiency. The corresponding $B$ values for each variant across different models are listed in Table 4.

Table 4: Model forward steps $B$ under different configurations.

| Model | Configuration | Model Forward Steps $B$ |
|---|---|---|
| Open-Sora 1.2 | Original | 30 |
| | LeMiCa-slow | 19 |
| | LeMiCa-fast | 11 |
| Latte | Original | 50 |
| | LeMiCa-slow | 27 |
| | LeMiCa-fast | 14 |
| CogVideoX | Original | 50 |
| | LeMiCa-slow | 27 |
| | LeMiCa-fast | 16 |

## C  OOD Generalization Analysis

We analyze the out-of-distribution (OOD) generalization ability of LeMiCa through two evaluation setups: VBench and IP-VBench.

**OOD Evaluation on VBench.** LeMiCa is evaluated on VBench [11], which follows a distribution distinct from T2V-CompBench [41] used for DAG construction. We quantify the distributional shift by computing prompt-level distances using text embeddings and PCA (see Table 5).

Table 5: Distributional distance analysis between VBench and T2V-CompBench.

| Attribute | Description | VBench |
|---|---|---|
| Distance | VBench vs. T2V-CompBench | 0.61 |
| Radius | 1 Std. Deviation of T2V-CompBench | 0.39 |
| Distance / Radius | Ratio of distance to radius | 1.58 |
| OOD Status* | Is it OOD? | ✓ |

Despite this evident OOD setting, LeMiCa consistently maintains strong acceleration and visual quality (Table 1), demonstrating robustness to unseen prompt distributions.

**OOD Evaluation on IP-VBench.** We further test LeMiCa on IP-VBench [1], which contains intentionally unrealistic and semantically diverse prompts across four domains: *Physical*, *Biological*, *Social*, and *Geographical*. These prompts differ significantly from training data, with Distance/Radius nearly doubling compared to T2V-CompBench (see Table 6).

Across all domains, LeMiCa substantially outperforms TeaCache in LPIPS, SSIM, and PSNR, underscoring its strong generalization and robustness under severe OOD conditions.

Table 6: Quantitative OOD performance on IP-VBench across four semantic domains.

| Method | Domain | LPIPS (↓) | SSIM (↑) | PSNR (↑) | Distance/Radius | OOD Status* |
|--------|--------|-----------|----------|----------|-----------------|-------------|
| TeaCache | Physical | 0.093 | 0.911 | 26.7 | 2.09 | ✓ |
| | Biological | 0.171 | 0.839 | 24.0 | 1.90 | ✓ |
| | Social | 0.144 | 0.842 | 24.9 | 1.93 | ✓ |
| | Geographical | 0.072 | 0.914 | 29.8 | 2.13 | ✓ |
| | **Overall** | **0.120** | **0.877** | **26.4** | **2.01** | ✓ |
| LeMiCa | Physical | 0.039 | 0.954 | 34.6 | 2.09 | ✓ |
| | Biological | 0.054 | 0.905 | 31.4 | 1.90 | ✓ |
| | Social | 0.040 | 0.946 | 33.1 | 1.93 | ✓ |
| | Geographical | 0.038 | 0.945 | 34.9 | 2.13 | ✓ |
| | **Overall** | **0.042** | **0.938** | **33.5** | **2.01** | ✓ |

## D   Offline Cost

The graph construction in LeMiCa is an entirely offline, three-stage process. First, Edge Weight Estimation estimates reconstruction errors by running full-generation passes on approximately 20 sampled prompts; this task is fully parallelizable (leveraging 8 GPUs in our experiments) and only needs to be run once per model configuration. The subsequent stages, Graph Construction (fusing jump edges into a sparse DAG) and Path Optimization (employing a lexicographic minimax search to find acceleration paths), are both highly efficient, each completing in under 1 second. As detailed in Table 7, these offline procedures incur negligible overhead, yet this low-cost offline computation yields up to **2.44×** acceleration during inference generation.

Table 7: Offline cost analysis of LeMiCa on OpenSora.

| Stage | Description | Time Cost (Ref) | Affects Inference |
|-------|-------------|-----------------|-------------------|
| Edge Weight Estimation | Full-generation error estimation | ∼3.18 min / prompt | No |
| Graph Construction | Build sparse DAG | <1 sec | No |
| Path Optimization | Minimax search for jump paths | <1 sec | No |
| Inference Acceleration | Execute jump paths with caching | Up to 2.44× faster | Yes |

## E   More Visual Results

We present additional visual comparisons across three foundational models: Open-Sora [53], Latte [24], and CogVideoX [45]. Results are grouped into two settings: fidelity-focused and speed-focused.

### E.1   Fidelity-Focused

We perform frame-by-frame comparisons to assess fine-grained differences in quality (LeMiCa-slow vs. TeaCache-slow). Since this setting uses relatively low acceleration ratios, artifacts are less obvious in real-time playback. To address this, we extract representative frames that highlight detail preservation, object integrity, and temporal consistency. As shown in Figures 8, 9, 10, 11, and 12, our method consistently produces more coherent results across all baselines.

### E.2   Speed-Focused

To evaluate robustness under aggressive acceleration, we compare videos generated with higher speed-up ratios (LeMiCa-fast vs. TeaCache-fast). This setting is designed to prioritize generation speed without significantly compromising visual quality. Under such conditions, baseline methods are more prone to issues such as flickering, object drift, and reduced temporal consistency. In contrast, our method maintains strong temporal and semantic coherence, even at high generation speeds.

As part of the supplementary material, we include the following video files: **Speed-Focused Open-Sora.mp4**, **Speed-Focused Latte.mp4**, and **Speed-Focused CogVideoX.mp4**.

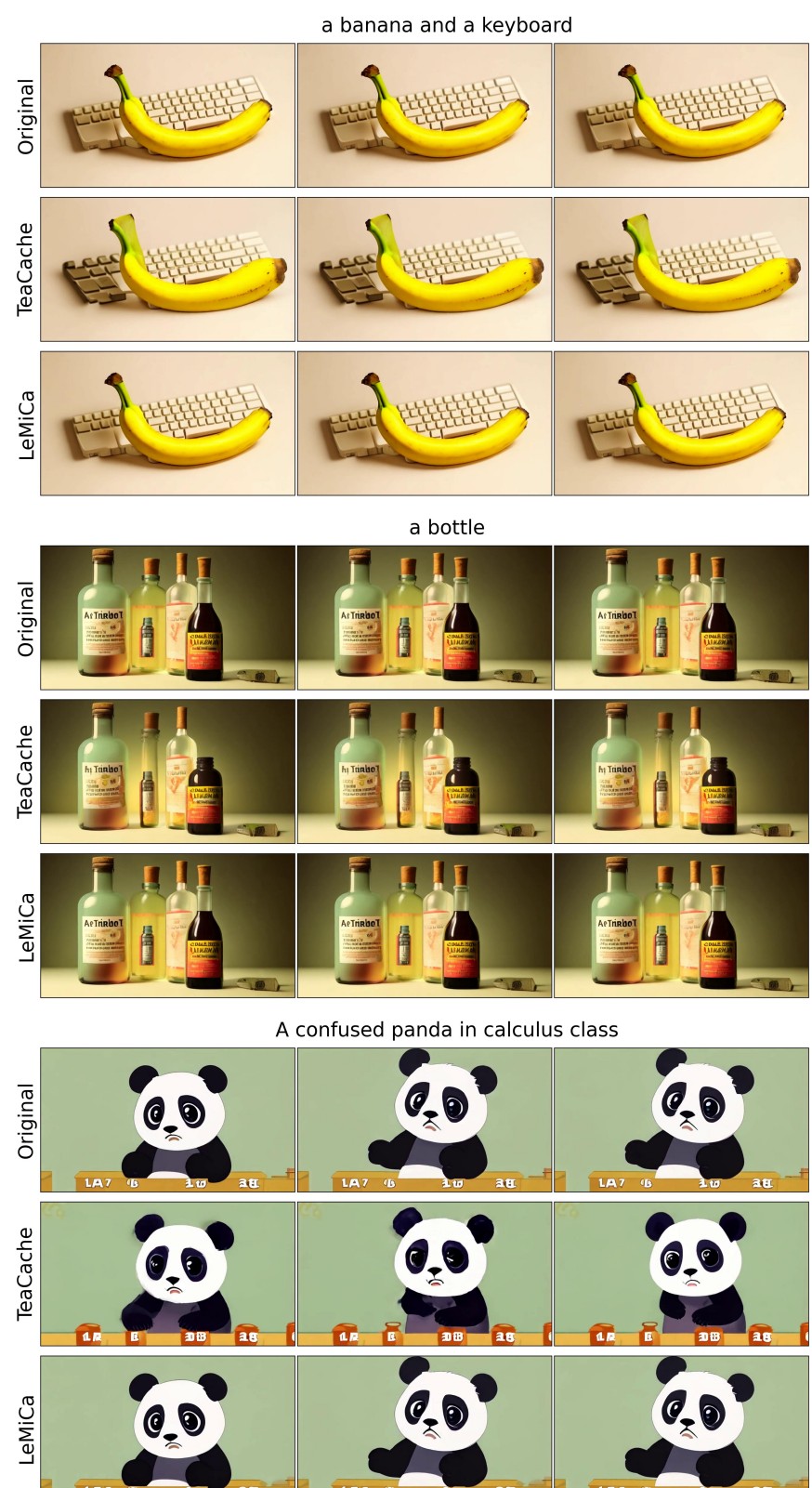

Figure 8: More visual results on Open-Sora (Part I).

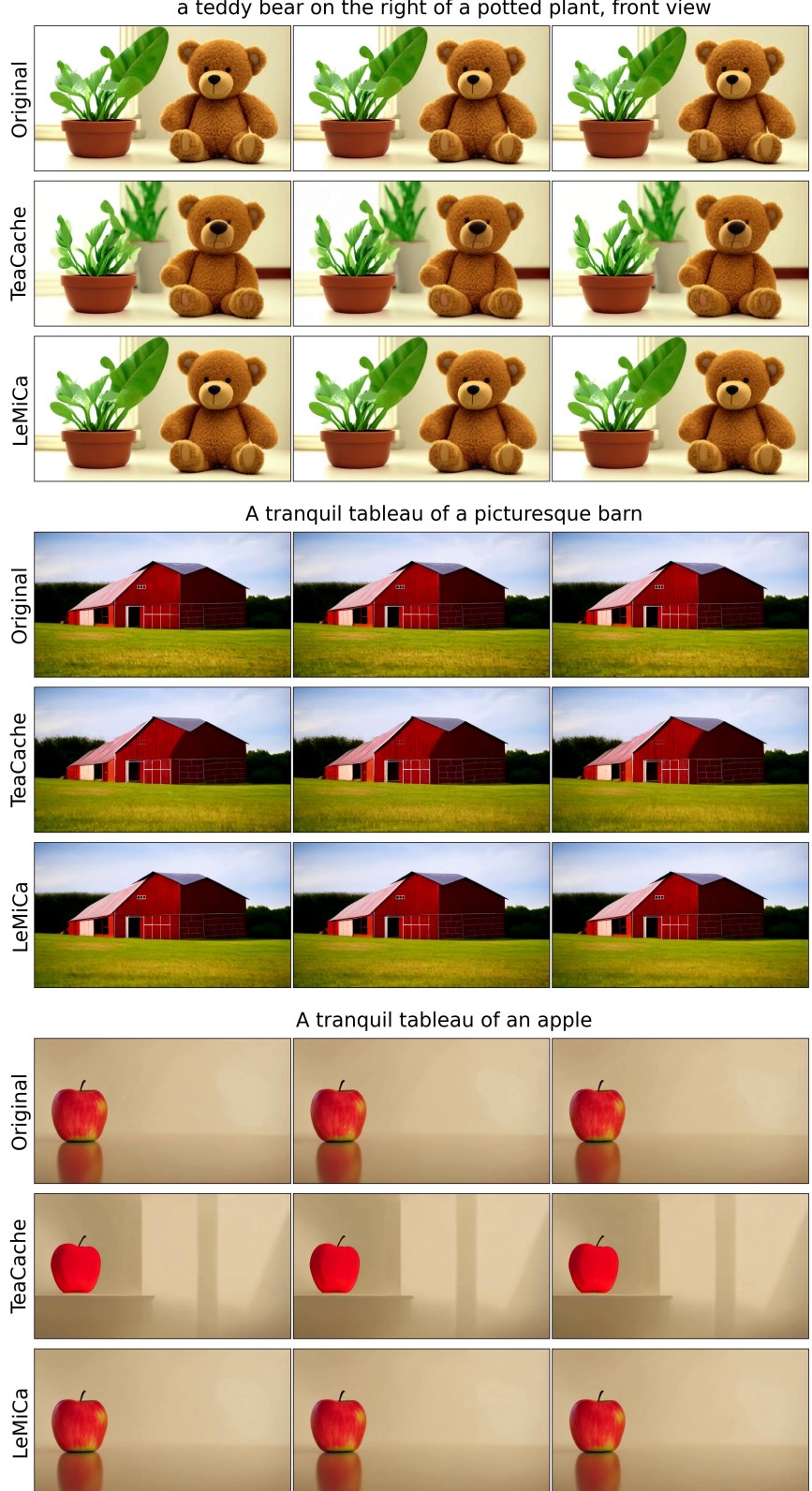

Figure 9: More visual results on Open-Sora (Part II).

An astronaut flying in space, zoom in

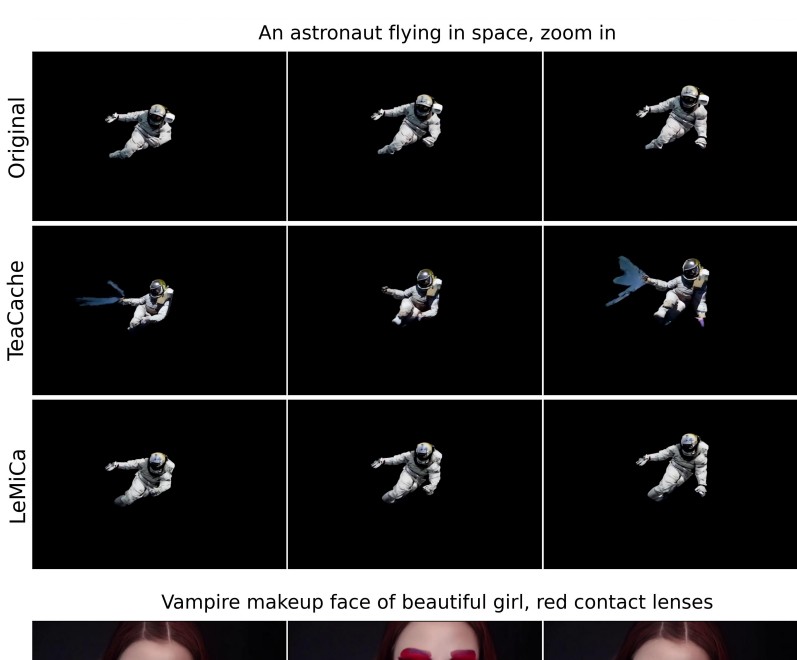

Vampire makeup face of beautiful girl, red contact lenses

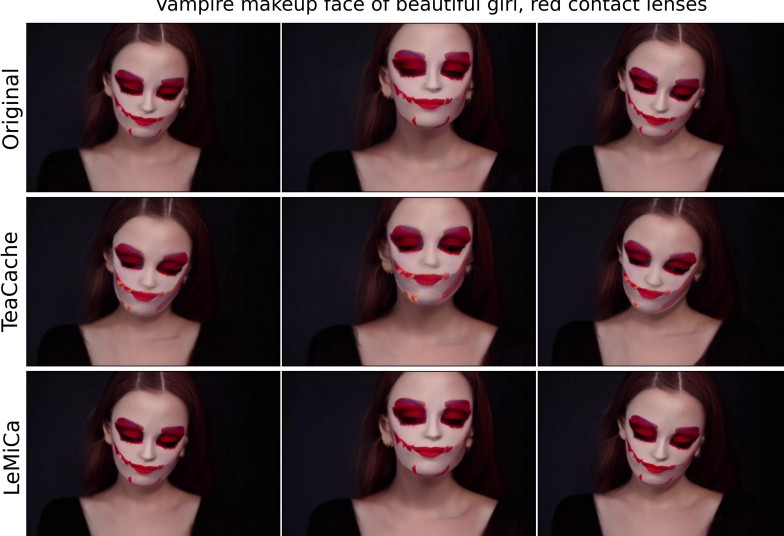

Gwen Stacy reading a book,
featuring a steady and smooth perspective

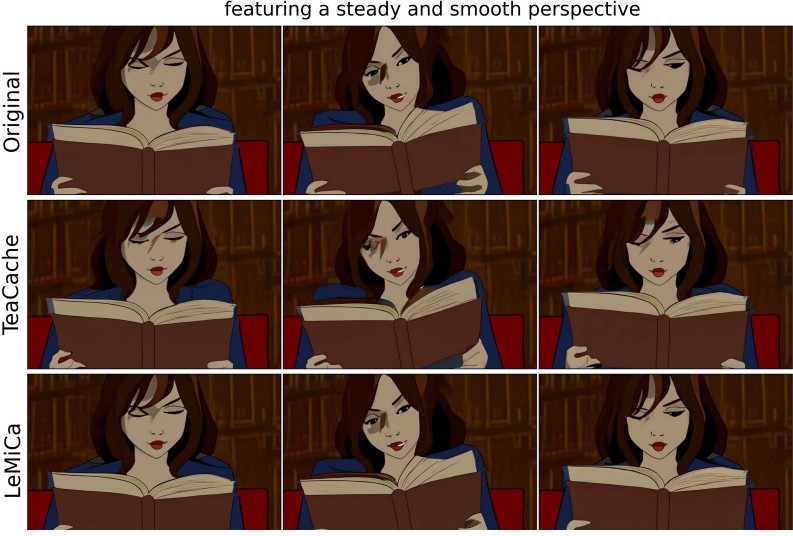

Figure 10: More visual results on CogVideoX.

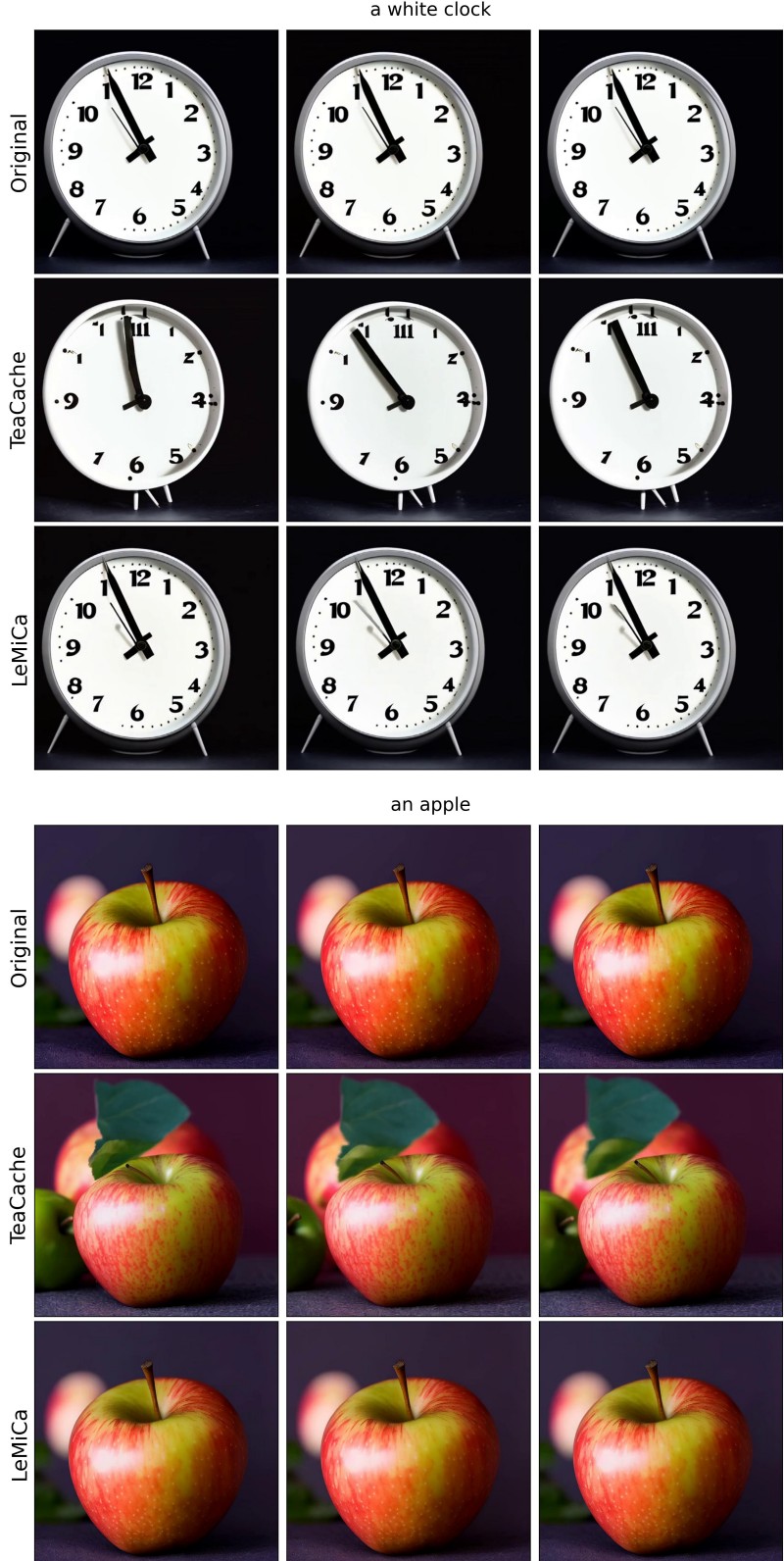

Figure 11: More visual results on Latte (Part I).

An astronaut is riding a horse in the space inaphotorealistic style

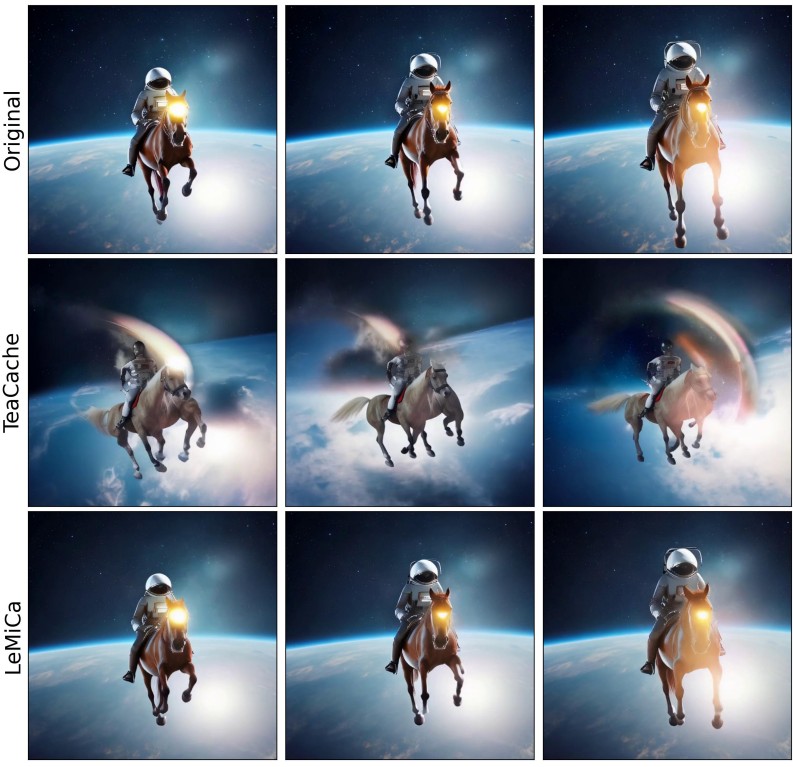

A boat sailing leisurely along the Seine River with the Eiffel Tower in background, surrealism style

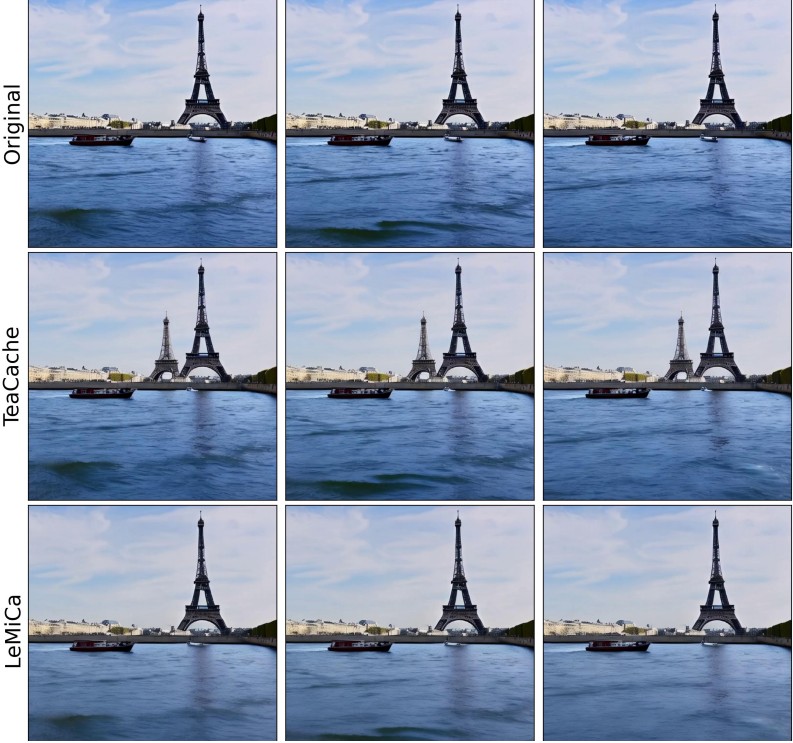

Figure 12: More visual results on Latte (Part II).

## F  Limitation

Although our method achieves strong performance in both acceleration and video fidelity, it still has certain limitations. First, when the original video quality is low, particularly in scenarios involving complex motion dynamics, it struggles to consistently generate satisfactory results. This reflects a dependency on the representational capacity of the underlying diffusion model. Second, under high acceleration ratios, some degree of quality degradation remains inevitable due to the significantly reduced number of model forward steps. We believe that continued progress in foundational video generation models will help alleviate these issues. Moreover, since our approach focuses solely on temporal step scheduling and is agnostic to model architecture, it can be quickly adapted to future, more powerful diffusion models.

## G  Social Impact

Diffusion-based video generation models are often limited by high inference time and computational cost. Our method alleviates this by significantly improving efficiency without requiring additional training. This enables broader access to high-quality video synthesis, particularly in resource-constrained settings. By reducing computation during the inference process, our approach also lowers energy use and carbon emissions, contributing to more sustainable AI development. Furthermore, we will release our code to support future research.

