# OpenReview forum: "LeMiCa: Lexicographic Minimax Path Caching for Efficient Diffusion-Based Video Generation"
_NeurIPS.cc/2025/Conference — NeurIPS 2025 spotlight_

### Official Review · Reviewer_Jva8 · 2025-06-29

**Clarity:** 3
**Significance:** 3
**Originality:** 3
**Rating:** 5
**Confidence:** 3

**Summary:**

This paper introduces LeMiCa, a novel, training-free, and model-agnostic framework for accelerating diffusion-based video generation. The key idea is to improve caching decisions by framing them as a global path planning problem using a Lexicographic Minimax Path Optimization approach. Rather than relying on local heuristics for cache selection, LeMiCa constructs a directed acyclic graph (DAG) where each edge represents a potential cache segment, weighted by its global error impact. This enables selection of caching paths that bound the worst-case degradation.

**Questions:**

1. How sensitive is LeMiCa to the choice of prompts used for DAG construction?
Although ablations are provided (e.g., 1–350 samples), can performance degrade under domain shifts, such as out-of-distribution prompts or styles?

2. Why choose lexicographic minimax over simpler bounded-error strategies?
Could a hybrid strategy (e.g., combining minimax with average-case bounding) perform better in certain quality-speed regimes?

3. What is the overhead of the offline DAG construction in practice?
Please provide empirical numbers (e.g., time, memory) for building the cache DAG on large models like CogVideoX.

4. Could LeMiCa generalize to adaptive or online caching?
The current setup is entirely static. In latency-constrained deployments (e.g., real-time rendering), could partial DAG updates or dynamic adjustments help?

5. Missing error bars/statistical significance.
Given the strong claims of performance gains, please provide error bars or confidence intervals over multiple seeds or trials. This would improve confidence in reported improvements and potentially increase the final score.

**Ethical Concerns:**

["NO or VERY MINOR ethics concerns only"]

**Final Justification:**

The authors have addressed my concerns by providing a complete rebuttal. I would be happy to raise my score.

**Limitations:**

Yes

**Quality:**

3

**Strengths And Weaknesses:**

Strengths:
1. Clear conceptual novelty: The formulation of caching as a global lexicographic minimax optimization problem is a well-justified and innovative leap from local greedy heuristics.
2. Strong empirical results: Extensive experiments across three T2V models consistently show LeMiCa's advantages in LPIPS, SSIM, PSNR, and VBench metrics.
3. Training-free and model-agnostic: The framework does not require architectural changes or fine-tuning, broadening its applicability.
4. Scalability and robustness: Effective across varying resolutions, frame counts, and denoising trajectories.
5. Well-organized experimental section, including ablations and robustness checks.

Weaknesses:
1. Error estimation remains heuristic: While global in scope, the edge weights used in the DAG are derived from averaging over prompts and may miss task-specific subtleties.
2. No runtime overhead claim may underplay preprocessing: The offline DAG construction phase is significant, especially for real-time applications.
3. Limited theoretical insight into optimization guarantees: The use of lexicographic minimax is intuitive, but theoretical convergence or optimality bounds are not discussed.
4. No uncertainty quantification: Error bars/statistical significance are missing from results, which limits rigor.

---

> ### Author Rebuttal · Authors · 2025-07-31
>
> We sincerely thank the reviewer for the insightful comment.
>
> > **W1: The globally constructed DAG may overlook task-specific nuances due to prompt-averaged edge weights.**
>
> We average errors across prompts when computing DAG edge weights for two key reasons:
>
> **Stability**: Prompt-specific errors can vary due to stochasticity, especially in early diffusion steps. Averaging reduces variance and yields more stable estimates.
>
> **Simplicity and generality**: Mean aggregation is model-agnostic, easy to compute, and aligns well with our lexicographic minimax path objective.
>
> While LeMiCa may not always yield the best path for every individual prompt, it generalizes well. As shown in our ablation (Line 203), it builds strong schedules with just 20 samples and performs effectively on unseen prompts and seeds.
>
> ---
>
> > **Q1: How sensitive is LeMiCa to the choice of prompts used for DAG construction? Can its performance degrade under domain shifts?**
>
> We appreciate the reviewer for raising a concern similar to those mentioned by the other two reviewers.
>
> LeMiCa is **not sensitive** to the specific prompts used for DAG construction.  Despite clear **out of distribution (OOD)** conditions, LeMiCa consistently delivers strong acceleration and maintains high generation quality.
>
> Our experiments support this conclusion across multiple dimensions:
> - **Cross-dataset generalization**: DAGs built on T2V-CompBench perform well on both VBench and IP-VBench, despite substantial distribution gaps.
> - **Sample Requirements**: LeMiCa constructs effective cache paths using as few as 20 samples.
> - **OOD Evaluation**: IP-VBench exhibits over a 2× standard deviation shift from T2V-CompBench, yet LeMiCa still significantly outperforms TeaCache, achieving an LPIPS of only 0.04.
>
> For detailed results and OOD analysis, please refer to our response to **Reviewer KfV5’s W1 & Q1** and **Reviewer iNvD’s W3**.
>
> ---
> > **Q2-1: Why choose lexicographic minimax over simpler bounded-error strategies?**
>
> We choose lexicographic minimax because it enables global optimization and directly controls the worst-case error, which is crucial for stable and high-quality video generation.
>
> In contrast, bounded-error strategies suffer from two key limitations:
>
> * **Temporal heterogeneity**: The error characteristics across different time steps in the denoising process vary significantly, making a single threshold inadequate for all situations.（Line 36）
> * **Local greediness**: Even with dynamic thresholds (e.g., TeaCache), bounded-error strategies still fall into local greedy errors. (Line 47)
>
> ---
> > **Q2-2: Could a hybrid strategy (e.g., combining minimax with average-case bounding) perform better in certain quality-speed regimes?**
>
> The idea of combining minimax with average-case bounding for path search in a DAG is intriguing, but we believe it may not yield better results in certain quality-speed regimes, for the following reasons:
>
> - **Offline Nature of Graph Construction and Path Search**
>   Both graph construction and path search are performed offline. As a result, any computational overhead introduced by a hybrid strategy would not affect the model\'s real-time inference speed, nor improve inference efficiency.
>
> - **Path Selection Comparison and Experiments**
>   We compared various path selection methods, including Shortest Path and MiniMax Path (Line 220). We also tested using statistical bounds, such as average error and variance, for path selection. However, our results consistently show that MiniMax Path is most effective at controlling worst-case errors.
>
> ---
>
> > **W2&Q3: Performance analysis over the cost of the graph construction.**
>
> In LeMiCa, the graph construction is performed entirely offline and consists of the following three stages:
>
> (1) **Edge Weight Estimation:** We estimate reconstruction errors of candidate jump edges by running full-generation passes on ~20 sampled prompts. This step is relatively time-consuming but fully parallelizable and only needs to be run once per model configuration.
>
> (2) **Graph Construction:** Valid jump edges are fused into a sparse DAG. Thanks to explicit constraints on skip length, the resulting structure is simple and this step completes in under 1 second.
>
> (3) **Path Optimization:** A lexicographic minimax search is performed over the sparse graph to identify acceleration paths that minimize the worst-case reconstruction error. This process is also highly efficient, completing in under 1 sec.
>
> The table below summarizes the offline cost measured on CogVideoX:
>
> | Stage                  | Time Cost (Ref)     | VRAM       |
> | ---------------------- | ------------------- | ---------- |
> | Edge Weight Estimation  | 9.74 min / prompt  | 18718 MiB  |
> | Graph Construction      | < 1 sec             |  /         |
> | Path Optimization       | < 1 sec             |   /        |
>
> OpenSora:
>
> | Stage                  | Time Cost (Ref)     | VRAM       |
> | ---------------------- | ------------------- | ---------- |
> | Edge Weight Estimation  | 3.18 min / prompt  | 21718 MiB  |
> | Graph Construction      | < 1 sec             |  /         |
> | Path Optimization       | < 1 sec             |   /        |
>
> Latte:
>
> | Stage                  | Time Cost (Ref)     | VRAM       |
> | ---------------------- | ------------------- | ---------- |
> | Edge Weight Estimation  | 2.57 min / prompt  | 15664 MiB  |
> | Graph Construction      | < 1 sec             |  /         |
> | Path Optimization       | < 1 sec             |   /        |
>
> ---
>
> > **W3: Theoretical convergence or optimality bounds are not discussed.**
>
> Here are the proof of theoretical convergence optimality bounds. The full details of the procedure can be found in the supplementary material as **Algorithm 1**  (Lexicographic Minimax Path Selection).
>
> **Theorem 1: Convergence Guarantee**
>
> **Statement**: Given a directed acyclic graph $G=(V,E)$, source $s$, target $t$, and budget $B$, Algorithm 1 terminates in finite steps.
>
> **Proof**:
>
> (1) **Finite State Space:** The DP table has $|V| × B$ entries, each representing a finite optimization subproblem.
>
> (2) **Monotonic Updates:** For any state $(v,k)$, the value $dp[v][k]$ can only improve
> (decrease lexicographically) during iterations.
>
> (3) **Bounded Iterations:** The algorithm runs exactly $B$ outer iterations, processing each edge at most once per iteration.
>
> ---
> **Optimality Bound Interpretation：**
>
> The core advantage of lexicographic minimax optimization lies in its explicit control of worst-case local errors, a principle rooted in robust control theory [1]. Unlike additive objectives, it prevents large errors from dominating global behavior.
>
> **Classical Shortest Path is Suboptimal in Our Context**
>
> Shortest-path algorithms (e.g., Dijkstra [2]) minimize total cost, but they fail here due to:
>
> - **Exponential Error Amplification**: As shown in Figure 2b, early-stage cache errors get amplified nonlinearly. A single high-error edge early in the trajectory can lead to total collapse, even if total additive error is low.
> - **Over-Aggressive Skipping**: Shortest-path often chooses long skips (fewer nodes) to reduce total cost. However, long-range cache reuse often incurs very high local error due to non-local denoising mismatches.
>
> **References:**
>
> [1] Robust and optimal control. 1996
>
> [2] A note on two problems in connexion with graphs. 2022
>
> ---
>
> > **Q4: Could LeMiCa generalize to adaptive or online caching?**
>
> Thank you for the thoughtful suggestions.
>
> (1) Adaptive or Online Caching:
> LeMiCa currently uses a static offline DAG for acceleration. Extending it to adaptive or online caching is non-trivial, especially under latency constraints where real-time updates are required. We agree this is an important direction and plan to investigate efficient dynamic scheduling strategies in future work.
>
> (2) Partial DAG Updates:
> We see promise in partial DAG updates as a practical compromise. By locally adjusting edge weights or dependencies—akin to low-rank adaptations like LoRA—we can reduce recomputation while preserving graph structure.
>
> Future Outlook:
> In real-time tasks (e.g., interactive rendering), where prompts or scenes evolve smoothly, temporal coherence can be exploited to incrementally update the DAG. We believe this opens a promising path for extending LeMiCa to dynamic, low-latency settings.
>
>
> > **W4&Q5: Missing error bars/statistical significance.**
>
> Thank you for your feedback. We have now included them, along with statistical significance indicators, in the main experimental table. These updates will be reflected in the revised version of the paper.
>
>
>
> | Dataset           | Method     | LPIPS ↓            | SSIM ↑            | PSNR ↑          |
> | ----------------- | ---------- | ------------------ | ----------------- | --------------- |
> | **Open-Sora 1.2** | Original   |  —                  | —                 | —               |
> |                   | TeaCache   |  0.134 ± 0.012      | 0.837 ± 0.024     | 23.50 ± 1.3     |
> |                   | **LeMiCa** |  **0.050 ± 0.010**  | **0.923 ± 0.022** | **31.32 ± 1.1** |
> | **Latte**         | Original   |  —                  | —                 | —               |
> |                   | TeaCache   |  0.195 ± 0.019      | 0.775 ± 0.014     | 21.52 ± 1.2     |
> |                   | **LeMiCa** |  **0.091 ± 0.012**  | **0.865 ± 0.024** | **27.65 ± 1.2** |
> | **CogVideoX**     | Original   |  —                  | —                 | —               |
> |                   | TeaCache   |  0.053 ± 0.0095     | 0.928 ± 0.014     | 31.07 ± 1.5     |
> |                   | **LeMiCa** |  **0.023 ± 0.0039** | **0.958 ± 0.011** | **35.93 ± 1.3** |
>
> **Paired *t*-tests**  confirm that the improvements in LPIPS, SSIM, and PSNR are statistically significant at the ***p* < 0.01** level.
>
>
> ***
>
> We are grateful again for your insightful comments and believe the above clarifications address your key concerns. Please let us know if any further elaboration is needed.
>
> — Authors

---

> > ### Comment · Reviewer_Jva8 · 2025-08-03
> >
> > Thank the authors for the very complete rebuttal. I do not have further questions.

---

> > > ### Author Response · Authors · 2025-08-04
> > >
> > > Dear reviewer Jva8,
> > >
> > > We sincerely appreciate your thoughtful consideration of our rebuttal. Thank you for your positive feedback and recognition of our work. Some of your insightful suggestions may further enhance our research in the future.
> > >
> > > Once again, thank you for your time and effort in reviewing our paper.
> > >
> > > — Authors

---

### Official Review · Reviewer_iNvD · 2025-07-02

**Clarity:** 3
**Significance:** 3
**Originality:** 3
**Rating:** 4
**Confidence:** 3

**Summary:**

The paper proposes LeMiCa, a training-free acceleration framework for diffusion-based video generation that improves upon existing caching methods. The authors identify that "local-greedy" caching strategies, which make decisions based on adjacent timesteps, fail to control the global accumulation of errors. To solve this, LeMiCa formulates the cache scheduling process as a path optimization problem on a Directed Acyclic Graph (DAG). Each edge in the DAG represents a potential cache segment, weighted by its global impact on the final generated video output quality. The framework then uses a Lexicographic Minimax optimization strategy to find the optimal path that minimizes the worst-case error under a fixed inference budget. Experiments on state-of-the-art models like Open-Sora and Latte show that LeMiCa achieves significant speedups of up to 2.9x while better preserving video quality better compared to prior methods.

**Questions:**

My questions are listed in the weaknesses section.

**Ethical Concerns:**

["NO or VERY MINOR ethics concerns only"]

**Final Justification:**

I maintain my borderline accept rating. I still believe the videos provided for evaluation are too limited and this concern was not addressed, but my concerns on the graph construction are addressed in the rebuttal. The overall work is impactful so I maintain my score.

**Limitations:**

yes.

**Paper Formatting Concerns:**

No formatting concerns.

**Quality:**

3

**Strengths And Weaknesses:**

Strengths:

-	The paper proposes a novel approach to the caching problem. Shifting from a local decision process to a global path optimization problem is a logical and effective idea. The formulation of cache scheduling as a lexicographic minimax problem on a DAG is also novel/interesting and addresses the issue of compounding errors.

-	The experimental results are thorough and convincing. LeMiCa consistently outperforms SOTA caching methods like TeaCache across multiple models (Open-Sora, Latte, CogVideoX) and metrics. Table 1 shows LeMiCa achieving a better trade-off between speed and quality, and the qualitative results in Figures 3 and 4 visually confirm its superior editing quality as do the examples in the supplementary materials.

-	The ablation studies are thorough.

Weaknesses:

-	Does the computational complexity of graph construction lead to any slowdown? Can a performance analysis over the cost of the graph construction also be done similar to the ablations provided on performance at different resolutions and lengths.

-	The number of video examples provided in the supplementary are very limited (only 3), which leads to the possibility of the framework not being effective across a broad set of videos and only effective at maintaining quality on a small subset or cherry-picked examples.

---

> ### Author Rebuttal · Authors · 2025-07-31
>
> We sincerely thank **Reviewer iNvD** for the thoughtful questions and constructive feedback. Below are our detailed responses.
>
> ---
>
> > **W1: Does the computational complexity of graph construction lead to any slowdown?**
>
> The graph construction in LeMiCa does **not introduce any slowdown** during model inference. It is performed entirely offline as a one-time preprocessing step. Once constructed, the Directed Acyclic Graph (DAG) is used to compute a static acceleration path, which guides the model’s forward passes without adding any runtime overhead.
>
> ---
>
> > **W2: Performance analysis over the cost of the graph construction.**
>
> In LeMiCa, the graph construction is performed entirely offline and consists of the following three stages:
>
> (1) **Edge Weight Estimation:** We estimate reconstruction errors of candidate jump edges by running full-generation passes on ~20 sampled prompts. This step is relatively time-consuming but fully parallelizable and only needs to be run once per model configuration.
>
> (2) **Graph Construction:** Valid jump edges are fused into a sparse DAG. Thanks to explicit constraints on skip length, the resulting structure is simple and this step completes in under 1 second.
>
> (3) **Path Optimization:** A lexicographic minimax search is performed over the sparse graph to identify acceleration paths that minimize the worst-case reconstruction error. This process is also highly efficient, completing in under 1 sec.
>
> The table below summarizes the offline cost measured on OpenSora:
>
> | **Stage**               | **Description**                              | **Time Cost (Ref)** | **Affects Inference** |
> |-------------------------|-----------------------------------------------|----------------------|------------------------|
> | Edge Weight Estimation  | Full-gen error estimation (~20 prompts)       | ~3.18 min / prompt   | No                     |
> | Graph Construction      | Build sparse DAG                              | < 1 sec              | No                     |
> | Path Optimization       | Minimax search for jump paths                 | < 1 sec              | No                     |
> | Inference Acceleration  | Execute jump paths with caching               | Up to 2.44× faster   | Yes                    |
>
>
>
> > **W3: Does LeMiCa generalize well to a broader range of videos beyond those shown in the supplementary?**
>
> Addressing LeMiCa’s generalization capability, we highlight the following:
>
> (1) **OOD Evaluation (VBench):** For DAG construction, we used **T2V-CompBench**[1] (Line 174), while evaluations were done on **VBench**[2] (Line 165). These two datasets originate from entirely different sources and distributions. LeMiCa’s strong performance on VBench actually has already confirms its robustness to unseen prompts and scenes.（See the table below for OOD analysis.）
>
> (2) **Further OOD Evaluation (IP-VBench):** We further evaluate LeMiCa on **IP-VBench**[3] with the same constructed DAG path , a dataset featuring intentionally non-realistic, ceative video prompts. Its distributional distance from T2V-CompBench is nearly **2 standard deviations**, representing a truly diverse OOD space.
>
> * **OOD Analysis:**
> We computed OOD status by extracting text embeddings, applying PCA for dimensionality reduction, and measuring prompt-level distances. Specifically, we compared T2V-CompBench against VBench and IP-Vbench. The results are as follows:
>
> | **Attribute**     | **Description**                       | **VBench** |**IP-Vbench** |
> | ----------------- | ------------------------------------- | ---------- |--------------|
> | Distance          | T2V-CompBench  vs. ~              | 0.61       |0.85         |
> | Radius            | 1 Standard Deviation of T2V-CompBench | 0.39       |0.42         |
> | Distance / Radius | Ratio of distance to radius           | 1.58   | 2.01         |
> | OOD Status\*      | Is it OOD?                            | ✓      | ✓            |
>
>
> **All datasets > 1.5× radius, confirming OOD status.*
>
> * **IP-VBench Results:**
>
> | **Model**   | **LPIPS (↓)** | **SSIM (↑)** | **PSNR (↑)** |
> |-------------|---------------|--------------|--------------|
> | TeaCache    | 0.12          | 0.87         | 26.4         |
> | LeMiCa      | 0.04          | 0.93         | 33.5         |
>
> The table demonstrates that LeMiCa maintains strong performance on this challenging and diverse dataset.
>
> **References:**
>
> [1] T2v-compbench: A comprehensive benchmark for compositional text-to-video generation CVPR 2025
>
> [2] Vbench: Comprehensive benchmark suite for video generative models. CVPR 2024
>
> [3] Impossible Videos. ICML 2025
>
>
> ***
>
> We are grateful again for your insightful comments and believe the above clarifications address your key concerns. Please let us know if any further elaboration is needed.
>
> — Authors

---

> > ### Author Response · Authors · 2025-08-04
> >
> > Dear reviewer iNvD,
> >
> > We would like to express our sincere gratitude for your thoughtful review, your recognition of our work, and the insightful feedback you provided.
> >
> > Thank you again for your time and effort in reviewing our paper.
> >
> > — Authors

---

> ### Comment · Reviewer_iNvD · 2025-08-05
>
> I maintain my rating. I still believe the videos provided for evaluation are too limited and this concern was not fully addressed (as videos are not allowed in the rebuttal so this is not held against the authors rebuttal), but my concerns on the graph construction are addressed in the rebuttal. The overall work is impactful so I maintain my score (which is already a positive score in favor of acceptance).

---

> > ### Author Response · Authors · 2025-08-06
> >
> > Dear Reviewer iNvD,
> >
> > We understand your concern about the limited video examples (supplementary materials), sincerely appreciate your thoughtful feedback, and respect your decision. We regret that we couldn't show more videos due to rebuttal policies. After the review, we will upload more video results on our project page. We warmly invite you to take a look, and we hope this can help address your concern.
> >
> > Thank you again for your time and support.
> >
> > — Authors

---

### Official Review · Reviewer_J7ti · 2025-07-02

**Clarity:** 2
**Significance:** 2
**Originality:** 3
**Rating:** 4
**Confidence:** 3

**Summary:**

The paper introduces a training‑free framework for accelerating diffusion‑based video generators. Instead of re‑using UNet/DiT activations with simple “local‐greedy” rules (e.g., TeaCache, PAB), LeMiCa does the following: 1. Defines a “global outcome‑aware” error: how much a whole segment of cached steps changes the final decoded video, averaged over several prompts/seeds; 2. Builds an offline Directed Acyclic Graph (DAG) whose edges are candidate cache segments weighted by that global error; 3. Searches under a fixed inference‑budget B the path that lexicographically minimizes the worst‑case edge weight (the minimax path). At run‑time only those B timesteps are fully computed; all others reuse cached features, incurring virtually no online overhead.

**Questions:**

N/A

**Ethical Concerns:**

["NO or VERY MINOR ethics concerns only"]

**Final Justification:**

The authors have addressed my concerns by providing a complete rebuttal. I would be happy to raise my score.

**Quality:**

2

**Strengths And Weaknesses:**

Strength:

-- The paper articulates a genuine failure mode visible in prior work, providing a clear diagnosis of “local‑greedy” caching weaknesses (temporal heterogeneity & error accumulation).

-- The formulation is novel: casting cache selection as a global path search with a lexicographic minimax objective.

-- The paper provides thorough ablations: effect of sample count for graph construction, minimax vs shortest path, trajectory scale, resolution & length scaling.


Weakness:

-- The biggest problem of this work is the lack of convincing qualitative results. Firstly, I can only play the open-sora video in the supplementary material, not the other two; secondly, only three examples are shown, where none of them has large, rigid motions, but rather static scene with small motions; thirdly, for the three simple examples, I still observe significant artifacts: several parts are blurred out, and there is significant motion artifacts, e.g. where a snowflake blends with the background. These examples are not convincing for me that the approach does not suffer from much quality degradation, especially when comparing with TeaCache.

---

> ### Author Rebuttal · Authors · 2025-07-31
>
> We sincerely apologize for the inconvenience caused by the failure to play the supplementary videos. We fully understand that this may have affected your experience evaluating our work, and we greatly appreciate you bringing this issue to our attention.
>
> ---
>
> > **W1: Only the open-sora video was playable; the other two failed to play.**
>
> Upon receiving this feedback, we immediately investigated the issue and found that it was likely due to **video encoding incompatibility**. While the videos played correctly on some systems, certain decoders may not support the original encoding.
>
> To address this, we have tested and confirmed two reliable solutions across **macOS, Linux, and Windows** platforms:
>
> 1. **Re-encoding using the H.264 codec:**
>    ```bash
>    ffmpeg -i input.mp4 -vcodec h264 output.mp4
> 2. Using **VLC media player**, a free, open-source, cross-platform player that supports a wide range of formats.
>
> Once again, we sincerely apologize for the inconvenience and thank you for your understanding and patience.
>
> ---
>
> > **W2: Only three examples are shown, with no large, rigid motions—just static scenes with small movements.**
>
> We understand the concern regarding the lack of large, rigid motion in the current supplementary videos. We have supplemented our evaluation with additional motion-related metrics from VBench as well as a new dataset containing extensive rigid motions.
>
> (1) **VBench Motion Subsets:** LeMiCa outperforms TeaCache in three motion-related sub-tests:
>
> | **Method**   | **Human Action (↑)** | **Motion Smoothness (↑)** | **Dynamic Degree (↑)** |
> |----------|---------------|--------------------|----------------|
> | TeaCache | 0.83          | 0.981              | 0.49           |
> | LeMiCa   | 0.87          | 0.986              | 0.51           |
>
> (2) **IP-VBench Evaluation:** We further include evaluations on IP-VBench [1], which contains strong, diverse motion scenes—again confirming LeMiCa\'s robustness.
>
> | **Method** | **LPIPS (↓)** | **SSIM (↑)** | **PSNR (↑)** |
> | --------- | ------------- | ------------ | ------------ |
> | TeaCache  | 0.12          | 0.87         | 26.4         |
> | LeMiCa    | 0.04          | 0.93         | 33.5         |
>
> Additionally, we agree that visual examples of such scenes would better highlight our method’s capabilities. We will include more examples with large, rigid motions on the project website upon release.
>
> **References:**
>
> [1] Vbench: Comprehensive benchmark suite for video generative models. CVPR 2024
>
> [2] Impossible Videos. ICML 2025
>
> ---
> > **W3: The examples shown still have significant artifacts.**
>
> Thank you for the careful observation. We\'d like to clarify an important distinction in our method design:
>
> **Motion Blur (artifacts) ≠ Algorithmic Failure**
>
> A key design principle of LeMiCa is to **preserve consistency** between the accelerated and original videos (Line 31). Many of the “artifacts” observed, such as snowflake blending or motion blur, are inherent to the original video and result from camera motion or depth-of-field effects. LeMiCa intentionally preserves these natural dynamics, rather than suppressing them. In contrast, in two other videos where no noticeable artifacts were present in the original, LeMiCa also shows no artifacts, while retaining more fine-grained details and scene integrity than TeaCache.
>
> **Better Preservation of Spatial Relationships and Semantic Content:**
>
> LeMiCa maintains spatial consistency more effectively than TeaCache. For example, snowflakes consistently appear in front of the house, preserving depth cues. In contrast, TeaCache often misplaces or prematurely removes snowflakes, disrupting the scene’s spatial coherence.
>
> We also conducted a detailed comparison of key semantic elements between the original video, TeaCache, and LeMiCa outputs. As shown in the table below, LeMiCa consistently retains more important visual content, even under challenging and dynamic scenarios:
>
> | **Semantic Element**                  | **Original** | **TeaCache** | **LeMiCa** |
> |--------------------------------------|--------------|--------------|------------|
> | The Broken Bridge in the Lake        | ✓            | x            | ✓          |
> | The House in the Bottom-Left Corner  | ✓            | x            | ✓          |
> | Christmas Tree                       | ✓            | x            | ✓          |
> | Snowflake Density                    | ✓            | x            | ✓          |
>
>
> ---
> > **W4: These examples are not convincing for me that the approach does not suffer from much quality degradation, especially when comparing with TeaCache:**
>
> we respectfully disagree with the view that LeMiCa degrades more than TeaCache, and we support our claim with:
>
> **Quantitative Evidence:** In VBench, LeMiCa-fast consistently outperforms TeaCache-fast in LPIPS, SSIM, and PSNR (see Table 1, Line 162).
>
> **Qualitative Evidence:** LeMiCa-fast better preserves semantic details such as the shark, laptop, and background objects. (Figure 4 and the supplementary videos)
>
> **Robustness Evidence:** In our ablation study (Line 196), even under 3.54× acceleration, LeMiCa remains more stable than TeaCache.
>
>
> ***
>
> Please let us know if additional clarification would be helpful. We truly appreciate your constructive feedback.
>
> — Authors

---

> > ### Comment · Reviewer_J7ti · 2025-08-02
> >
> > Thank the authors for the very complete rebuttal. I do not have further questions and would like to raise my score towards a positive side.

---

> > > ### Author Response · Authors · 2025-08-04
> > >
> > > Dear reviewer J7ti,
> > >
> > > We sincerely appreciate your careful consideration of our rebuttal and your willingness to raise the score. Thank you for your positive evaluation and support.
> > >
> > > Thanks again for the time and effort in reviewing our work.
> > >
> > > — Authors

---

### Official Review · Reviewer_KfV5 · 2025-07-11

**Clarity:** 3
**Significance:** 2
**Originality:** 3
**Rating:** 4
**Confidence:** 3

**Summary:**

LeMiCa (Lexicographic Minimax Path Caching) is a new training-free acceleration method for text-to-video diffusion models. The core idea is to reuse intermediate model outputs (caching) while carefully controlling errors on a global scale. Unlike prior “local” caching heuristics that decide frame by frame, LeMiCa treats the whole generation process as a path through a directed acyclic graph (DAG) of diffusion timesteps. Each possible skip (cache segment) is an edge weighted by its impact on final output quality. The authors introduce a lexicographic minimax optimization to choose a caching path that minimizes the worst-case generation error under a fixed computation budget. This strategy explicitly bounds the largest error between the accelerated video and a full-quality video, preventing cumulative drift. In experiments on multiple video generation benchmarks, LeMiCa achieves significant speed-ups (e.g. ~2.9× faster on the Latte model) while maintaining high fidelity (e.g. LPIPS ≈0.05 on Open-Sora), outperforming previous caching techniques in both efficiency and visual consistency. The method requires no model retraining or architectural changes, highlighting its practical utility for faster diffusion-based video synthesis.

**Questions:**

- **Generality to New Content:** The approach builds a static error-weighted DAG using multiple prompts to estimate segment errors. How sensitive is this graph to the choice of calibration prompts or data domain? In other words, if one were to apply LeMiCa to very different videos or prompts not seen during graph construction, would the caching decisions remain effective? It would be helpful to discuss the method’s robustness to domain shifts or how one might update the cache strategy for new types of content.
- **Combination with Other Accelerations:** Since LeMiCa is *training-free* and model-agnostic, have the authors considered combining it with other inference acceleration techniques (e.g., model distillation or quantization)? For instance, could one first apply LeMiCa to reduce steps and also use a distilled model for larger jumps, or would that risk compounding errors? A brief discussion on how this caching strategy might complement (or differ from) training-based accelerations could clarify its place in the broader landscape of efficient diffusion methods.

**Ethical Concerns:**

["NO or VERY MINOR ethics concerns only"]

**Final Justification:**

I appreciate the authors' response and will maintain my rating as borderline accept.

**Limitations:**

No. The submission does not explicitly include a dedicated *limitations* section or discuss potential **societal impacts** of the work.

**Quality:**

3

**Strengths And Weaknesses:**

**Strengths:**

- **Practical Impact:** LeMiCa offers a *model-agnostic, training-free* acceleration framework that can be applied to existing diffusion video generators without retraining. This makes it highly practical for real-world use, enabling much faster inference (2–3× speed-ups in tested cases) while introducing only minimal perceptual degradation. Such immediate efficiency gains will be valuable for deploying diffusion models in interactive or production settings.
- **Robust Generation Quality:** The proposed lexicographic minimax strategy effectively controls error accumulation, leading to improved global consistency across frames. The results show that accelerated videos remain very close to originals in structure and detail (e.g., extremely low LPIPS error of 0.05), addressing a key weakness of prior caching methods that often caused content drift or loss of fine details. Notably, the paper demonstrates better structural alignment and style consistency than the state-of-the-art TeaCache baseline, indicating strong preservation of video quality even at high acceleration.
- **Thorough Evaluation:** The submission includes extensive experiments on multiple text-to-video benchmarks and base models (such as Latte and Open-Sora). It consistently shows **dual improvements** – faster inference and higher generation fidelity – across the board. Ablation studies and analyses of error metrics are provided, giving credibility to the approach’s advantages. The empirical results are convincing and reinforce the significance of the proposed method.

**Weaknesses:**

- **Complexity and Generalization:** LeMiCa’s global DAG construction involves evaluating error impacts with multiple prompts and selecting an optimal path. This added complexity means implementing the method requires extra analysis offline. The authors should clarify how general the precomputed graph is, for instance, does it reliably cover arbitrary prompts and scenes, or could certain out-of-distribution videos see different error dynamics? This is a minor issue, but discussing the method’s generalization beyond the tested data would strengthen the work.

---

> ### Author Rebuttal · Authors · 2025-07-31
>
> We extend our gratitude for your insightful feedback and suggestions.
>
> ***
>
> >**W1&Q1:DAG construction sensitivity and OOD generalization**
>
> Thank you for the insightful and closely related questions. We provide a unified response structured around three dimensions:
>
> (1) **Unified DAG construction:**
> For each model, LeMiCa constructs a single global DAG using just 4~70 prompts sampled from T2V-CompBench[1] (Line 174). This graph remains fixed across all evaluations, and performance saturates with these prompts.
>
> (2) **OOD Evaluation (VBench):**
> In our primary evaluation (Table 1), we assess LeMiCa’s effectiveness on VBench[2], which is actually a completely different distribution than T2V-CompBench. We quantified the distributional shift by computing prompt-level distances using text embeddings and PCA.
>
> | **Attribute**     | **Description**                       | **VBench** |
> | ----------------- | ------------------------------------- | ---------- |
> | Distance          | VBench vs. T2V-CompBench              | 0.61       |
> | Radius            | 1 Standard Deviation of T2V-CompBench | 0.39       |
> | Distance / Radius | Ratio of distance to radius           | **1.58**   |
> | OOD Status\*      | Is it OOD?                            | **✓**      |
>
> Despite the clear OOD status, LeMiCa consistently shows strong acceleration and quality maintenance on VBench.（Table 1, line161）
>
> (3) **Further OOD Evaluation (IP-VBench):**
> To test LeMiCa under more extreme OOD conditions, we evaluate on IPVBench[3], which contains intentionally unrealistic and unnatural prompts across four domains: Physical, Biological, Social, and Geographical. These prompts differ not only statistically but also conceptually from training data（Distance/Radiu nearly 2×）
>
> | **Method** | **IP-Vbench** | **LPIPS (↓)** | **SSIM (↑)** | **PSNR (↑)** | **Distance/Radius** | **OOD Status*** |
> | --------- | ------------- | ------------- | ------------ | ------------ | ------------------- | --------------- |
> | TeaCache  | Physical      | 0.093         | 0.911        | 26.7         | 2.09                | ✓               |
> |           | Biological    | 0.171         | 0.839        | 24.0         | 1.90                | ✓               |
> |           | Social        | 0.144         | 0.842        | 24.9         | 1.93                | ✓               |
> |           | Geographical  | 0.072         | 0.914        | 29.8         | 2.13                | ✓               |
> |           | **Overall**   | **0.12**      | **0.877**    | **26.4**     | **2.01**            | ✓               |
> | LeMiCa    | Physical      | 0.039         | 0.954        | 34.6         | 2.09                | ✓               |
> |           | Biological    | 0.054         | 0.905        | 31.4         | 1.90                | ✓               |
> |           | Social        | 0.040         | 0.946        | 33.1         | 1.93                | ✓               |
> |           | Geographical  | 0.038         | 0.945        | 34.9         | 2.13                | ✓               |
> |           | **Overall**   | **0.042**     | **0.938**    | **33.5**     | **2.01**            | ✓               |
>
> Despite the severity, LeMiCa consistently and significantly outperforms the Teacache baseline on all metrics, showing strong robustness in abstract and diverse scenarios.
>
> **References:**
>
> [1] T2v-compbench: A comprehensive benchmark for compositional text-to-video generation CVPR 2025
>
> [2] Vbench: Comprehensive benchmark suite for video generative models. CVPR 2024
>
> [3] Impossible Videos. ICML 2025
> ***
>
> > **Q2: Combined with distillation or quantization methods?**
>
> Thank you for the insightful suggestion. LeMiCa is fully compatible with other acceleration techniques such as distillation and quantization.
>
> (1) **Inference with Distilled Models:** We applied LeMiCa to a distilled 8-step model (WAN2.1-T2V within the DCM[1] framework) and further reduced it to 6 steps. Despite the model already being heavily compressed via distillation, LeMiCa brought additional acceleration with minimal quality degradation:
>
> | **Method**           | **NFE** | **LPIPS↓** | **SSIM↑** | **PSNR↑** |
> |---------------------|--------:|-----------|-----------|-----------|
> | DCM                 | 8       | —         | —         | —         |
> | DCM + TeaCache      | 6       | 0.29      | 0.63      | 14.07     |
> | **DCM + LeMiCa**    | 6       | **0.08**  | **0.87**  | **22.04** |
>
> This suggests LeMiCa can provide further gains even on top of state-of-the-art distillation methods, highlighting its strong complementarity in the inference phase.
>
> (2) **Other Possibilities (Distillation Training, Quantization):** LeMiCa’s training-free nature also makes it a potential companion to the distillation process itself.  Although we have not empirically evaluated this combination yet, their compatibility is clear in theory and suggests potential for additive speedups. Use LeMiCa to accelerate teacher model inference during training (e.g., for generating supervision signals) is an interesting direction that could reduce training costs. We consider this a promising avenue for future exploration.
>
> Additionally, For quantization model, LeMiCa are orthogonal to them in principle: the former reduces the number of steps, while the latter reduces the cost per step. Though we have not yet tested this combination, they are theoretically compatible and could offer additive speedups.
>
> **References:**
>
> [1] DCM: Dual-Expert Consistency Model for Efficient and High-Quality Video Generation 2025
>
> ***
>
> Please let us know if additional clarification would be helpful. We truly appreciate your constructive feedback.
>
> — Authors

---

### Note · Authors · 2025-08-12

Dear Area Chair and Reviewers,

We sincerely thank the reviewers for their constructive feedback, which has helped us clarify and strengthen our work.

**1. Core Contribution**

LeMiCa’s key idea is to overcome the **local-greedy trap** in prior caching by formulating it as a **global graph-based optimization problem** and introducing a lexicographic minimax strategy for DAG optimization, which achieves faster inference and better generation quality.

We appreciate all reviewers’ unanimous recognition of this idea:

> “Addressing a key weakness of prior caching methods that often caused content drift or loss of fine details.” — R-KfV5

> “The paper articulates a genuine failure mode visible in prior work, providing a clear diagnosis of local-greedy caching weaknesses.” — R-J7ti

> “Shifting from a local decision process to a global path optimization problem is a logical and effective idea.” — R-iNvD

> “Clear conceptual novelty: The formulation of caching as a global lexicographic minimax optimization problem is a well-justified and innovative leap from local greedy heuristics.” — R-Jva8

**2. Clarifications on Concerns**

- **Videos (supplementary material):** Two supplementary videos had playback issues due to encoding; we provided two alternative solutions to resolve this (thanks to R-J7ti). For the limited number of videos, due to rebuttal policy—though keyframe comparisons are already included—we will release more after the review (thanks to R-iNvD).
- **OOD evaluation:** We clarified that the graph-construction and test datasets differ in distribution and added results on IP-VBench, a more challenging OOD dataset, which further confirms LeMiCa’s strong generalization (R-KfV5, R-iNvD, R-Jva8).

**3. Revisions Made**
- Added error bars and statistical significance in Table 1 (R-Jva8).
- Highlighted dataset distribution differences in L175–L176 and L204–L206, and included OOD evaluation in Appendix B.4 (R-KfV5, R-iNvD, R-Jva8).
- Added a clearer description of graph construction cost in L173, with detailed experiments in Appendix B.5 (R-iNvD, R-Jva8).
- Prepared more video examples (to be released after the review).

We believe these clarifications and revisions address the reviewers’ concerns. We appreciate the constructive feedback and are encouraged by the indications of score increases. We look forward to these contributions supporting the AC–reviewer discussion and the final decision.

Sincerely,
Authors

---

### Decision · Program_Chairs · 2025-09-17

**Decision:**

Accept (spotlight)

**Comment:**

AC finds the paper interesting where a DAG is constructed offline with some prompts which is then used to guide the caching for speedup. What immediately comes to mind is whether this offline method, constructed with a few prompts, is generalizable and whether the prompts could cause a big difference. Reviewers pointed out the same concerns. There were also concerns with the quality of the videos in the supp.

After rebuttal, we see that the authors provided extra experiments on OOD which convinced the reviewers. The AC felt that more scientific insight can be provided why the method would generalize and why only a few prompts are needed, in addition to the new experimental results, which will make the paper stronger. Authors also clarified that the video quality is bottlenecked by the original model's performance, which is acceptable.

Overall, AC thinks the biggest strength of the paper is its novelty in using a DAG to conduct caching that results in a more global effect, which outweighs the concerns, and decides to accept the paper.